# Symbolic Physics Learner: Discovering governing equations via Monte Carlo tree search

**Fangzheng Sun[1], Yang Liu[2], Jian-Xun Wang[3], Hao Sun[4,*]**

[1]Northeastern University, Boston, MA, USA; [2]University of Chinese Academy of Sciences, Beijing, China;
[3]University of Notre Dame, Notre Dame, IN, USA; [4]Renmin University of China, Beijing, China.

Emails: *sun.fa@northeastern.edu*; *liuyang22@ucas.ac.cn*; *jwang33@nd.edu*; *haosun@ruc.edu.cn*

## ABSTRACT

Nonlinear dynamics is ubiquitous in nature and commonly seen in various science and engineering disciplines. Distilling analytical expressions that govern nonlinear dynamics from limited data remains vital but challenging. To tackle this fundamental issue, we propose a novel Symbolic Physics Learner (SPL) machine to discover the mathematical structure of nonlinear dynamics. The key concept is to interpret mathematical operations and system state variables by computational rules and symbols, establish symbolic reasoning of mathematical formulas via expression trees, and employ a Monte Carlo tree search (MCTS) agent to explore optimal expression trees based on measurement data. The MCTS agent obtains an optimistic selection policy through the traversal of expression trees, featuring the one that maps to the arithmetic expression of underlying physics. Salient features of the proposed framework include search flexibility and enforcement of parsimony for discovered equations. The efficacy and superiority of the SPL machine are demonstrated by numerical examples, compared with state-of-the-art baselines.

## 1 INTRODUCTION

We usually learn the behavior of a nonlinear dynamical system through its nonlinear governing differential equations. These equations can be formulated as $\dot{\mathbf{y}}(t) = d\mathbf{y}/dt = \mathcal{F}(\mathbf{y}(t))$, where $\mathbf{y}(t) = \{y_1(t), y_2(t), ..., y_n(t)\} \in \mathbb{R}^{1 \times n_s}$ denotes the system state at time $t$, $\mathcal{F}(\cdot)$ a nonlinear function set defining the state motions and $n_s$ the system dimension. The explicit form of $\mathcal{F}(\cdot)$ for some nonlinear dynamics remains underexplored. For example, in a mounted double pendulum system, the mathematical description of the underlying physics might be unclear due to unknown viscous and frictional damping forms. These uncertainties yield critical demands for the discovery of nonlinear dynamics given observational data. Nevertheless, distilling the analytical form of governing equations from limited noisy data, commonly seen in practice, is an intractable challenge.

Ever since the early work on the data-driven discovery of nonlinear dynamics (Džeroski & Todorovski, 1993; Dzeroski & Todorovski, 1995), many scientists have stepped into this field of study. During the recent decade, the escalating advances in machine learning, data science, and computing power have enabled several milestone efforts of unearthing the governing equations for nonlinear dynamical systems. Notably, a breakthrough model named SINDy (Sparse Identification of Nonlinear Dynamics) (Brunton et al., 2016) has shed light on tackling this achallenge. SINDy was invented to determine the sparse solution among a pre-defined basis function library recursively through a sequential threshold ridge regression (STRidge) algorithm. SINDy quickly became one of the state-of-art methods and kindled significant enthusiasm in this field of study (Rudy et al., 2017; Long et al., 2018; Champion et al., 2019; Chen et al., 2021; Sun et al., 2021; Rao et al., 2022). However, the success of this sparsity-promoting approach relies on a properly defined candidate function library that requires good prior knowledge of the system. It is also restricted by the fact that a linear combination of candidate functions might be insufficient to recover complicated mathematical expressions. Moreover, when the library size is massive, it empirically fails to hold the sparsity constraint.

At the same time, attempts have been made to tackle the nonlinear dynamics discovery problems by introducing neural networks with activation functions replaced by commonly seen mathematical

---

*Corresponding author

operators (Martius & Lampert, 2017; Sahoo et al., 2018; Kim et al., 2019; Long et al., 2019). The intricate formulas are obtained via symbolic expansion of the well-trained network. This interpretation of physical laws results in larger candidate pools compared with the library-based representation of physics employed by SINDy. Nevertheless, since the sparsity of discovered expressions is primarily achieved by empirical pruning of the network weights, this framework exhibits sensitivity to user-defined thresholds and may fall short to produce parsimonious equations for noisy and scarce data.

Alternatively, another inspiring work (Bongard & Lipson, 2007; Schmidt & Lipson, 2009) re-envisioned the data-driven nonlinear dynamics discovery tasks by casting them into symbolic regression problems which have been profoundly resolved by the genetic programming (GP) approach (Koza & Koza, 1992; Billard & Diday, 2003). Under this framework, a symbolic regressor is established to identify the governing equations that best describe the underlying physics through free combination of mathematical operators and symbols, leading to great flexibility in model selection. One essential weakness of this early methodology is that, driven exclusively by the goal of empirically seeking the best-fitting expression (e.g. minimizing the mean-square error) in a genetic expansion process, the GP-based model usually over-fits the target system with numerous false-positive terms under data noise, even sometimes at a subtle level, causing huge instability and uncertainty. However, this ingenious idea has inspired a series of subsequent endeavors (Cornforth & Lipson, 2012; Gaucel et al., 2014; Ly & Lipson, 2012; Quade et al., 2016; Vaddireddy et al., 2020). In a more recent work, Deep Symbolic Regression (DSR) (Petersen et al., 2021; Mundhenk et al., 2021), a reinforcement learning-based model was established and generally outperformed the GP based models including the commercial Eureqa software (Langdon & Gustafson, 2010). Additionally, the AI-Feynman methods (Udrescu & Tegmark, 2020; Udrescu et al., 2020; Udrescu & Tegmark, 2021) ameliorated symbolic regression for distilling physics laws from data by combining neural network fitting with a suite of physics-inspired techniques. This approach is also highlighted by a recursive decomposition of a complicated mathematical expression into different parts on a tree-based graph, which disentangles the original problem and speeds up the discovery. It outperformed Eureqa in the uncovering Feynman physics equations (Feynman et al., 1965). However, this approach is built upon ad-hoc steps and, to some extent, lacks flexible automation in equation discovery.

The popularity of adopting the tree-based symbolic reasoning of mathematical formulas (Lample & Charton, 2019) has been rising recently to discover unknown mathematical expressions with a reinforcement learning agent (Kubalík et al., 2019; Petersen et al., 2021; Mundhenk et al., 2021). However, some former work attempting to apply the Monte Carlo tree search (MCTS) algorithm as an alternative to GP for symbolic regression (Cazenave, 2013; White et al., 2015; Islam et al., 2018; Lu et al., 2021) failed to leverage the full flexibility of this algorithm, resulting in the similar shortage that GP-based symbolic regressors possess as discussed earlier. Despite these outcomes, we are conscious of the strengths of the MCTS algorithm in equation discovery: it enables the flexible representation of search space with customized computational grammars to guide the search tree expansion. A sound mathematical underpinning for the trade-off between exploration and exploitation is remarkably advantageous as well. These features make it possible to inform the MCTS agent by our prior physics knowledge in nonlinear dynamics discovery rather than randomly searching in large spaces.

**Contribution.** We propose a promising model named Symbolic Physics Learner (SPL) machine, empowered by MCTS, for discovery of nonlinear dynamics. This architecture relies on a grammar composed of (i) computational rules and symbols to guide the search tree spanning and (ii) a composite objective rewarding function to simultaneously evaluate the generated equations with observational data and enforce the sparsity of the expression. Moreover, we design multiple adjustments to the conventional MCTS by: **(1)** replacing the expected reward in UCT score with maximum reward to better fit the equation discovery objective, **(2)** employing an adaptive scaling in policy evaluation which would eliminate the uncertainty of the reward value range owing to the unknown error of the system state derivatives, and **(3)** transplanting modules with high returns to the subsequent search as a single leaf node. With these adjustments, the SPL machine is capable of efficiently uncovering the best path to formulate the complex governing equations of the target dynamical system.

## 2 BACKGROUND

In this section, we expand and explain the background concepts brought up in the introduction to the SPL architecture, including the expression tree (parse tree) and the MCTS algorithm.

**Expression tree.** Any mathematical expression can be represented by a combinatorial set of symbols and mathematical operations, and further expressed by a parse tree structure (Hopcroft et al., 2006; Kusner et al., 2017) empowered by a context-free grammar (CFG). A CFG is a formal grammar characterized by a tuple comprised of 4 elements, namely, $\mathcal{G} = (V, \Sigma, R, S)$, where $V$ denotes a finite set of non-terminal nodes, $\Sigma$ a finite set of terminal nodes, $R$ a finite set of production rules, each interpreted as a mapping from a single non-terminal symbol in $V$ to one or multiple terminal/non-terminal node(s) in $(V \cup \Sigma)^*$ where $*$ represents the Kleene star operation, and $S$ a single non-terminal node standing for a start symbol. In our work, equations are symbolized into parse trees: we define the start node as equation symbol $f$, terminal symbols (leaf nodes) corresponding to the independent variables formulating the equation (e.g., $x$, $y$), and a *placeholder symbol* $C$ for identifiable constant coefficients that stick to specific production rules. The non-terminal nodes between root and leaf nodes are represented by some symbols distinct from the start and terminal nodes (i.e., $M$). The production rules denote the commonly seen mathematical operators: unary rules (one non-terminal node mapping to one node) for operators like $\cos(\cdot)$, $\exp(\cdot)$, $\log(|\cdot|)$, and binary rules (one non-terminal node mapping to two nodes) for operators such as $+, -, \times, \div$. A parse tree is then generated via a pre-order traversal of production rules rooted at $f$ and terminates when all leaf nodes are entirely filled with terminal symbols. Each mathematical expression can be represented by such a traversal set of production rules.

**Monte Carlo tree search.** Monte Carlo tree search (MCTS) (Coulom, 2006) is an algorithm for searching optimal decisions in large combinatorial spaces represented by search trees. This technique complies with the best-first search principle based on the evaluations of stochastic simulations. It has already been widely employed in and proved the spectacular success by various gaming artificial intelligence systems, including the famous AlphaGo and AlphaZero (Silver et al., 2017) for computer Go game. A basic MCTS algorithm is composed of an iterative process with four steps:

1. **Selection.** The MCTS agent, starting from the root node, moves through the visited nodes of the search tree and selects the next node according to a given selection policy until it reaches an expandable node or a leaf node.

2. **Expansion.** At an expandable node, the MCTS agent expands the search tree by selecting one of its unvisited children.

3. **Simulation.** After expansion, if the current node is non-terminal, the agent performs one or multiple independent simulations starting from the current node until reaching the terminal state. In this process, actions are randomly selected.

4. **Backpropagation.** Statistics of nodes along the path from the current node to the root are updated with respect to search results (scores evaluated from the terminate states reached).

To maintain a proper balance between the less-tested paths and the best policy identified so far, the MCTS agent sticks to a trade-off between exploration and exploitation by taking action that maximizes the Upper Confidence Bounds applied for Trees (UCT), formulated as (Kocsis & Szepesvári, 2006):

$$UCT(s, a) = Q(s, a) + c\sqrt{\ln[N(s)]/N(s, a)} \tag{1}$$

where $Q(s, a)$ is the average result/reward of playing action $a$ in state $s$ in the simulations performed in the history, encouraging the exploitation of current best child node; $N(s)$ is number of times state $s$ visited, $N(s, a)$ the number of times action $a$ has been selected at state $s$, and $\sqrt{\ln[N(s)]/N(s, a)}$ consequently encourages exploration of less-visited child nodes. Constant $c$ controls the balance between exploration and exploitation, empirically defined upon the specific problem. Theoretical analysis of UCT-based MCTS (e.g., convergence, guarantees) is referred to Shah et al. (2019).

## 3 METHODS

Existing studies show that the MCTS agent continuously gains knowledge of specified tasks via the expansion of the search tree and, based on the backpropagation of evaluation results (i.e., rewards and number of visits), render a proper selection policy on visited states to guide the upcoming searching (Silver et al., 2017). In the proposed SPL machine, such a process is integrated with the symbolic reasoning of mathematical expressions to reproduce and evaluate valid mathematical expressions of the physical laws in nonlinear dynamics step-by-step, and then obtain a favorable selection policy

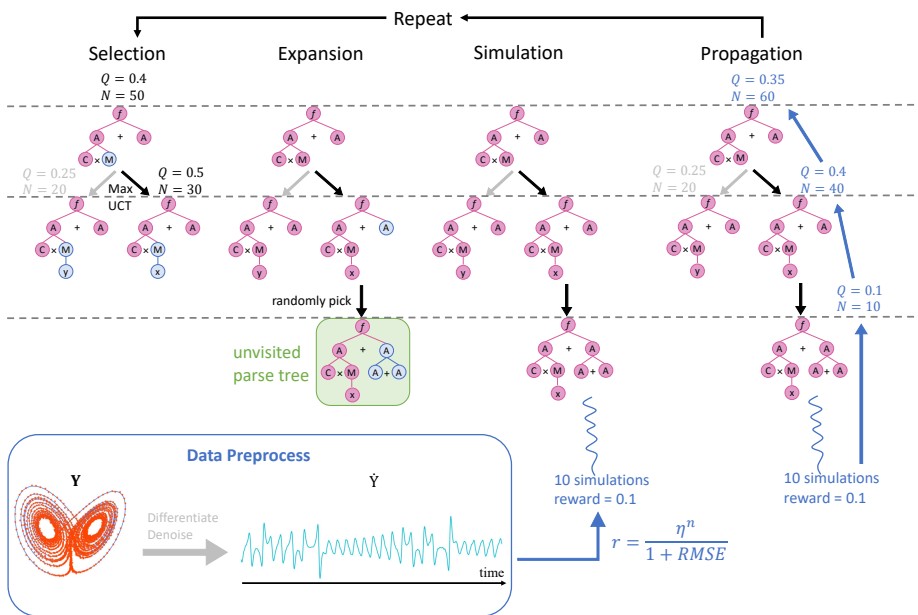

Figure 1: Schematic architecture of the SPL machine for nonlinear dynamics discovery. The graph explains the 4 MCTS phases of one learning episode with an illustrative example.

pointing to the best solution. This algorithm is depicted in Figure 1 with an illustrative example and its overall training scheme is shown in Algorithm 1. Discussion of the hyperparameter setting for this algorithm is given in Appendix Section A.

**Rewarding.** To evaluate the mathematical expression $\tilde{f}$ projected from a parse tree, we define a numerical reward $r \in \mathcal{R} \subset \mathbb{R}$ based on this expression and input data $\mathcal{D} = \{\mathbf{Y}; \dot{Y}_i\}$, serving as the search result of the current expansion or simulation. It is formulated as

$$r = \frac{\eta^n}{1 + \sqrt{\frac{1}{N} \left\| \dot{Y}_i - \tilde{f}(\mathbf{Y}) \right\|_2^2}} \tag{2}$$

where $\mathbf{Y} = \{\mathbf{y}_1, \mathbf{y}_2, ..., \mathbf{y}_m\} \in \mathbb{R}^{m \times N}$ is the $m$ dimensional state variables of a dynamical system, $\dot{Y}_i \in \mathbb{R}^{1 \times N}$ the numerically estimated state derivative for $i$th dimension, and $N$ the number of measurement data points. $\eta$ denotes a discount factor, assigned slightly smaller than 1; $n$ is empirically defined as the total number of production rules in the parse tree. This numerator arrangement is designated to penalize non-parsimonious solutions. This rewarding formulation outputs a reasonable assessment to the distilled equations and encourages parsimonious solution by discounting the reward of a non-parsimonious one. The rooted mean square error (RMSE) in denominator evaluates the goodness-of-fit of the discovered equation w.r.t. the measurement data.

**Training scheme.** A grammar $\mathcal{G} = (V, \Sigma, R, S)$ is defined with appropriate nodes and production rules to cover all possible forms of equations. To keep track of non-terminal nodes of the parsing tree, we apply a last-in-first-out (LIFO) strategy and denote the non-terminal node placed last on the stack $NT$ as the current node. We define the action space $\mathcal{A} = R$ and the state space $\mathcal{S}$ as all possible traversals of complete/incomplete parse trees (i.e., production rules selected) in ordered sequences. At the current state $s_t = [a_1, a_2, ...a_t]$ where $t \in \mathbb{N}$ is the discrete traversal step-index of the upcoming production rule, the MCTS agent masks out the invalid production rules for current non-terminal node and on that basis selects a valid rule as action $a_{t+1}$ (i.e, the left-hand side of a valid production rule is the current non-terminal symbol). Consequently, the parse tree gains a new terminal/non-terminal branch in accordance with $a_{t+1}$, meanwhile the agent finds itself in a new state $s_{t+1} = [a_1, a_2, ...a_t, a_{t+1}]$. The agent subsequently pops off the current non-terminal symbol from $NT$ and pushes the non-terminal nodes, if there are any, on the right-hand side of the selected rule onto the stack. Once the agent attains an unvisited node, a certain amount of simulations are performed, where the agent starts to randomly select the next node until the parse tree is completed.

---

**Algorithm 1:** Training SPL for discovering the $i^{th}$ governing equation ($i = 1, 2, ..., m$)

---

1   **Input:** Grammar $G = (V, \Sigma, R, S)$, measurement data $\mathcal{D} = \{\mathbf{Y}; \dot{Y}_i\}$;
2   **Parameters:** discount/regularization factor $\eta$, exploration rate $c$, $t_{max}$;     # $\eta$ controls equation parsimony;
3   **Output:** Optimal governing equation $\tilde{f}^\star$;
4   **for** *each episode* **do**
5      **Selection:** Initialize $s_0 = \emptyset, t = 0, NT = [S]$;
6      **while** $s_t$ *expandable and* $t < t_{max}$ **do**
7         Choose $a_{t+1} = \arg\max_{\mathcal{A}} UCT(s_t, a)$;
8         Take action $a_{t+1}$, observe $s'$, $NT$;
9         $s_{t+1} \leftarrow s'$ note as visited, $t \leftarrow t + 1$;
10      **end**
11      **Expansion:** Randomly take an unvisited path with action $a$, observe $s'$, $NT$;
12      $s_{t+1} \leftarrow s'$ note as visited, $t \leftarrow t + 1$;
13      **if** $NT = \emptyset$ **then**
14         Project $\tilde{f}$, **Backpropagate** $r_{t+1}$ and visited count and finish the episode;
15      **end**
16      **Simulation:** Fix the starting point $s_t, NT$;
17      **for** *each simulation* **do**
18         **while** $s_t$ *non-terminal and* $t < t_{max}$ **do**
19            Randomly take an action $a$, observe $s'$, $NT$;
20            $s_{t+1} \leftarrow s', t \leftarrow t + 1$;
21         **end**
22         **if** $NT = \emptyset$ **then**
23            Project $\tilde{f}$ and calculate $r_{t+1}$;
24         **end**
25      **end**
26      **Backpropagate** simulation results;
27   **end**

---

The reward is calculated or the maximal size is exceeded, resulting in a zero reward. The best result from the attempts counts as the reward of the current simulation phase and backpropagates from the current unvisited node all the way to the root node.

**Greedy search.** Different from the MCTS-based gaming AIs where the agents are inclined to pick the action with a high expected reward (average returns), the SPL machine seeks the unique optimal solution. In the proposed training framework, we apply a greedy search heuristic to encourage the agent to explore the branch which yields the best solution in the past: $Q(s, a)$ is defined as the maximum reward of the state-action pair, and its value is backpropagated from the highest reward in the simulations upon the selection of the pair. Meanwhile, to overcome the local minima problems due to this greedy approach in policy search, we enforce a certain level of randomness by empirically adopting the $\epsilon$-greedy algorithm, a commonly seen approach in reinforcement learning models.

**Adaptive-scaled rewarding.** Owing to the unknown level of error from the numerically estimated state derivatives, the range of the RMSE in the SPL reward function is unpredictable. This uncertainty affects the scale of rewarding values thus the balance between exploration and exploitation is presented in Eq. (1). Besides adding "1" to the denominator of Eq. (2) to avoid dramatically large numerical rewards, we also apply an adaptive scale of the reward, given by

$$Q(s, a) = \frac{r^*(s, a)}{\max_{s' \in \mathcal{S}, a' \in \mathcal{A}} Q(s', a')} \tag{3}$$

where $r^*$ denotes the maximum reward of the state-action pair. It is scaled by the current maximum reward among all states $\mathcal{S}$ and actions $\mathcal{A}$ to reach an equilibrium that the $Q$-values are stretched to the scale $[0, 1]$ at any time. This self-adaptive fashion yields a well-scaled calculation of UCT under different value ranges of rewards throughout the training.

**Module transplantation.** A function can be decomposed into smaller modules where each is simpler than the original one (Udrescu et al., 2020). This modularity feature, as shown in Figure 2, helps us develop a divide-and-conquer heuristic for distilling some complicated functions: after every certain amount of MCTS iterations, the discovered parse trees with high rewards are picked out and thereupon

reckoned as individual production rules and appended to the set of production rules $R$; accordingly, these trees are "transplanted" to the future ones as their modules (i.e, the leaves). To avoid early overfitting problem, we incrementally enlarge the sizes of such modules from a baseline length

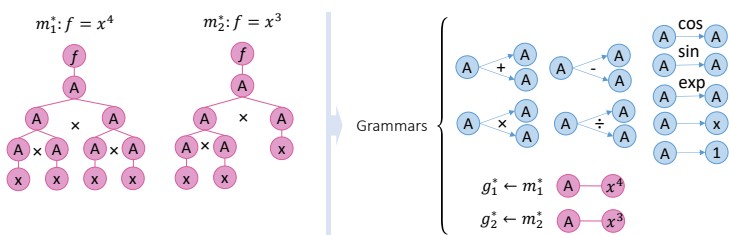

Figure 2: A module transplantation process: A complete parse serves as a single production rule and is appended to the grammar pool.

to the maximum allowed size of the parse tree throughout the iterations. The augmentation part of $R$ is refreshed whenever new production rules are created, keeping only the ones engendering high rewards. This approach accelerates the policy search by capturing and locking some modules that likely contribute to, or appear as part of the optimal solution, especially in the cases of the mathematical expression containing "deep" operations (e.g., high-order polynomials) whose structures are difficult for the MCTS agent to repeatedly obtain during the selection and expansion.

# 4 SYMBOLIC REGRESSION: FINDING MATHEMATICAL FORMULAS

## 4.1 DATA NOISE & SCARCITY

Data scarcity and noise are commonly seen in measurement data and become one of the bottleneck issues for discovering the governing equations of nonlinear dynamics. Tackling the challenges in high-level data scarcity and noise situations is traditionally regarded as an essential robustness indicator for a nonlinear dynamics discovery model. To this end, we present an examination of the proposed SPL machine by an equation discovery task in the presence of multiple levels of data noise and

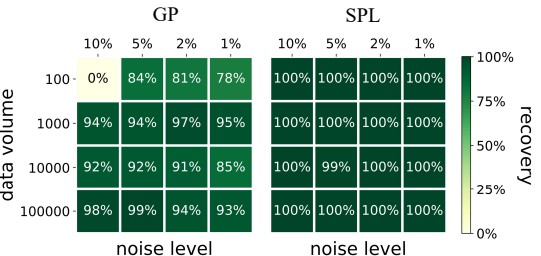

Figure 3: The effect of data noise/scarcity on recovery rate. The heatmaps demonstrate the recovery rate of GP and the SPL machine under different data conditions, summarized over 100 independent trials.

volume, comparing with a GP-based symbolic regressor (implemented with *gplearn* python package)[1]. The target equation is $f(x) = 0.3x^3 + 0.5x^2 + 2x$, and the independent variable $X$ is uniformly sampled in the given range $[-10, 10]$. Gaussian white noise is added to the dependent variable $Y$ with the noise level defined as the root-mean-square ratio between the noise and the exact values. For discovery, the two models are fed with equivalent search space: $\{+, -, \times, \div, cost, x\}$ as candidate mathematical operations and symbols. The hyperparameters of the SPL machine are set as $\eta = 0.99$, $t_{max} = 50$, and 10,000 episodes of training is regarded as one trail. For the GP-based symbolic regressor, the population of programs is set as 2,000, the number of generations as 20. The range of constant coefficient values is $[-10, 10]$. For 16 different data noise and scarcity levels, each model was performed 100 independent trails. The recovery rates are displayed as a $4 \times 4$ mesh grid w.r.t. different noise/scarcity levels in Figure 3. It is observed that the SPL machine outperforms the GP-based symbolic regressor in all the cases. A T-test also proves that the recovery rate of the SPL machine is significantly higher than that of GP (e.g., $p$-value $= 1.06 \times 10^{-7}$).

## 4.2 NGUYEN'S SYMBOLIC REGRESSION BENCHMARK

Nguyen's symbolic regression benchmark task (Uy et al., 2011) is widely used to test the model's robustness in symbolic regression problems. Given a set of allowed operators, a target equation, and data generated by the specified equation (see Table 1 for example), the tested model is supposed to distill the mathematical expression that is identical to the target equation, or equivalent to it (e.g., Nguyen-7 equation can be recovered as $\log(x^3 + x^2 + x + 1)$, Nguyen-10 equation can be recovered as $\sin(x + y)$, and Nguyen-11 equation can be recovered as $\exp(y\log(x))$). Some variants of Nguyen's benchmark equations are also considered in this experiment. Their discoveries require numerical

---

[1]All simulations are performed on a standard workstation with a NVIDIA GeForce RTX 2080Ti GPU.

Table 1: Recovery rate of three algorithms in Nguyen's benchmark symbolic regression problems. The SPL machine outperforms the other two models in average recovery rate.

| Benchmark | Expression | SPL | NGGP | GP |
|---|---|---|---|---|
| Nguyen-1 | $x^3 + x^2 + x$ | 100% | 100% | 99% |
| Nguyen-2 | $x^4 + x^3 + x^2 + x$ | 100% | 100% | 90% |
| Nguyen-3 | $x^5 + x^4 + x^3 + x^2 + x$ | 100% | 100% | 34% |
| Nguyen-4 | $x^6 + x^5 + x^4 + x^3 + x^2 + x$ | 99% | 100% | 54% |
| Nguyen-5 | $\sin(x^2)\cos(x) - 1$ | 95% | 80% | 12% |
| Nguyen-6 | $\sin(x^2) + \sin(x + x^2)$ | 100% | 100% | 11% |
| Nguyen-7 | $\ln(x + 1) + \ln(x^2 + 1)$ | 100% | 100% | 17% |
| Nguyen-8 | $\sqrt{x}$ | 100% | 100% | 100% |
| Nguyen-9 | $\sin(x) + \sin(y^2)$ | 100% | 100% | 76% |
| Nguyen-10 | $2\sin(x)\cos(y)$ | 100% | 100% | 86% |
| Nguyen-11 | $x^y$ | 100% | 100% | 13% |
| Nguyen-12 | $x^4 - x^3 + \frac{1}{2}y^2 - y$ | 28% | 4% | 0% |
| Nguyen-1$^c$ | $3.39x^3 + 2.12x^2 + 1.78x$ | 100% | 100% | 0% |
| Nguyen-2$^c$ | $0.48x^4 + 3.39x^3 + 2.12x^2 + 1.78x$ | 94% | 100% | 0% |
| Nguyen-5$^c$ | $\sin(x^2)\cos(x) - 0.75$ | 95% | 98% | 1% |
| Nguyen-8$^c$ | $\sqrt{1.23x}$ | 100% | 100% | 56% |
| Nguyen-9$^c$ | $\sin(1.5x) + \sin(0.5y^2)$ | 96% | 90% | 0% |
| Average | | **94.5%** | 92.4% | 38.2% |

estimation of the constant coefficient values. Each equation generates two datasets: one for training and another for testing. The discovered equation that perfectly fits the testing data is regarded as a successful discovery (i.e., the discovered equation should be identical or equivalent to the target one). The recovery rate is calculated based on 100 independent tests for each task. In these benchmark tasks, three algorithms are tested: GP-based symbolic regressor, the neural-guided GP (NGGP) (Mundhenk et al., 2021), and the SPL machine. Note that NGGP is an improved approach over DSR (Petersen et al., 2021). They are given the same set of candidate operations: $\{+, -, \times, \div, \exp(\cdot), \cos(\cdot), \sin(\cdot)\}$ for all benchmarks and $\{\sqrt{\cdot}, \ln(\cdot)\}$ are added to the 7, 8, 11 benchmarks. The hyperparameters of the GP-based symbolic regressor are the same as those mentioned in Section 4.1; configurations of the NGGP models are obtained from its source code[2]; detailed setting of the benchmark tasks and the SPL model is described in Appendix Section B. The success rates are shown in Table 1. It is observed that the SPL machine and the NGGP model both produce reliable results in Nguyen's benchmark problems and the SPL machine slightly outperforms NGGP. This experiment betokens the capacity of the SPL machine in discovery of equations with divergent forms.

**Ablation Study:** We consider four ablation studies by removing: (a) the adaptive scaling in reward calculation, (b) the discount factor $\eta^n$ that drives equation parsimony in Eq. (2), (c) module transplantation in tree generation, and (d) all of the above. The four models were tested on the first 12 Nguyen equations (see Appendix Section C). Results show the average recovery rates for these models are all smaller than that produced by SPL (see Appendix Table C.1), where the module transplantation brings the largest effect. Hence, these modules are critical to guarantee the proposed model efficacy.

## 5 PHYSICAL LAW DISCOVERY: FREE FALLING BALLS WITH AIR RESISTANCE

It is well known that, in 1589–1592, Galileo dropped two objects of unequal mass from the Leaning Tower of Pisa and drew a conclusion that their velocities were not affected by the mass. This has been well recognized globally as the "textbook" physical law for the vertical motion of a free-falling object:

Table 2: Baseline models ($c_i$: unknown constants).

| Physics Model | Derived model expression |
|---|---|
| Model-1 | $H(t) = c_0 + c_1 t + c_2 t^2 + c_3 t^3$ |
| Model-2 | $H(t) = c_0 + c_1 t + c_2 e^{c_3 t}$ |
| Model-3 | $H(t) = c_0 + c_1 \log(\cosh(c_2 t))$ |

[2]NGGP source code: https://github.com/brendenpetersen/deep-symbolic-optimization/tree/master/dso/dso

the height of the object is formulated as $H(t) = h_0 + v_0 t - \frac{1}{2} g t^2$, where $h_0$ denotes initial height, $v_0$ the initial velocity, and $g$ the gravitational acceleration. However, this ideal situation is rarely reached in our daily life because air resistance serves as a significant damping factor that prevents the above physical law from occurring in real-life cases.

Many efforts have been made to uncover the effect of the air resistance and derive mathematical models to describe the free-falling objects with air resistance (Clancy, 1975; Lindemuth, 1971; Greenwood et al., 1986). This section provides data-driven discovery of the physical laws of relationships between height and time in the cases of free-falling objects with air resistance based on multiple experimental ball-drop datasets (de Silva et al., 2020), which contain the records of 11 different types of balls dropped from a

Table 3: Mean square error (MSE) between ball motion prediction with the measurements in the test set. The SPL machine reaches the best prediction results in most (9 out of 11) cases.

| Type | SPL | Model-1 | Model-2 | Model-3 |
|------|-----|---------|---------|---------|
| baseball | **0.3** | 2.798 | 94.589 | 3.507 |
| blue basketball | **0.457** | 0.513 | 69.209 | 2.227 |
| green basketball | **0.088** | 0.1 | 85.435 | 1.604 |
| volleyball | **0.111** | 0.574 | 80.965 | 0.76 |
| bowling ball | **0.003** | 0.33 | 87.02 | 3.167 |
| golf ball | **0.009** | 0.214 | 86.093 | 1.684 |
| tennis ball | **0.091** | 0.246 | 72.278 | 0.161 |
| whiffle ball 1 | 1.58 | 1.619 | 65.426 | **0.21** |
| whiffle ball 2 | **0.099** | 0.628 | 58.533 | 0.966 |
| yellow whiffle ball | **0.428** | 17.341 | 44.984 | 2.57 |
| orange whiffle ball | 0.745 | **0.379** | 36.765 | 3.257 |

bridge (see Appendix Figure D.1). For discovery, each dataset is split into a training set (records from the first 2 seconds) and a testing set (records after 2 seconds). Three mathematically derived physics models are selected from the literature as baseline models[3,4,5] for this experiment (see Table 2), and the unknown constant coefficient values are estimated by Powell's conjugate direction method (Powell, 1964). Based on our prior knowledge of the physical law that may appear in this case, we use $\{+, -, \times, \div, \exp(\cdot), \cosh(\cdot), \log(\cdot)\}$ as the candidate grammars for the SPL discovery, with terminal nodes $\{t, const\}$. The hyperparameters are set as $\eta = 0.9999$, $t_{max} = 20$, and one single discovery is built upon 6,000 episodes of training. The physical laws distilled by SPL from training data are applied to the test data and compared with the ground truth. Their prediction errors, in terms of MSE, are presented in Table 3 (the SPL-discovered equations are shown in Appendix Table D.1). The full results can be found in Appendix Section D. It can be concluded that the data-driven discovery of physical laws leads to a better approximation of the free-falling objects with air resistance.

## 6 CHAOTIC DYNAMICS DISCOVERY: THE LORENZ SYSTEM

The first nonlinear dynamics discovery example is a 3-dimensional Lorenz system (Lorenz, 1963) whose dynamical behavior $(x, y, z)$ is governed by $\dot{x} = \sigma(y - x), \dot{y} = x(\rho - z) - y, \dot{z} = xy - \beta z$ with parameters $\sigma = 10$, $\beta = 8/3$, and $\rho = 28$. The Lorenz attractor has two lobes and the system, starting from anywhere, makes cycles around one lobe before switching to the other and iterates repeatedly, exhibiting strong chaos. The synthetic system states $(x, y, z)$ are generated by solving the nonlinear differential equations using the Matlab `ode113` (Shampine, 1975; Shampine & Reichelt, 1997) function. 5% Gaussian white noise is added to the clean data to generate noisy measurement.

Table 4: Summary of the discovered governing equations for Lonrez system. Each cell concludes if target physics terms are distilled (if yes, number of false positive terms in uncovered expression).

| Model | $\dot{x}$ | $\dot{y}$ | $\dot{z}$ |
|-------|-----------|-----------|-----------|
| Eureqa | Yes (1) | Yes (3) | Yes (1) |
| pySINDy | Yes (1) | No (N/A) | Yes (2) |
| NGGP | Yes (10) | Yes (8) | Yes (16) |
| SPL | **Yes (0)** | **Yes (0)** | **Yes (0)** |

The derivatives of the system states $(\dot{x}, \dot{y}, \dot{z})$ are unmeasured but estimated by central difference and smoothed by the Savitzky–Golay filter (Savitzky & Golay, 1964) in order to reduce the noise effect.

In this experiment, the proposed SPL machine is compared with three benchmark methods: Eureqa, pySINDy and NGGP. For Eureqa, NGGP, and the SPL machine, $\{+, -, \times, \div\}$ are used as candidate operations; the upper bound of complexity is set to be 50; for pySINDy, the candidate function library includes all polynomial basis of $(x, y, z)$ from degree 1 to degree 4. Appendix Table E.1

---

[3]Model 1: https://faraday.physics.utoronto.ca/IYearLab/Intros/FreeFall/FreeFall.html
[4]Model 2: https://physics.csuchico.edu/kagan/204A/lecturenotes/Section15.pdf
[5]Model 3: https://en.wikipedia.org/wiki/Free_fall

presents the distilled governing equations by each approach and Table 4 summarizes these results: the SPL machine uncovers the explicit form of equations accurately in the context of active terms, whereas Eureqa, pySINDy and NGGP yield several false-positive terms in the governing equations. In particular, although Eureqa and NGGP are capable of uncovering the correct terms, their performance is very sensitive to the measurement noise as indicated by the redundant terms (despite with small coefficients) shown in Appendix Table E.1. Overall, the baseline methods fail to handle the large noise effect, essentially limiting their applicability in nonlinear dynamics discovery. It is evident that the SPL machine is capable of distilling the concise symbolic combination of operators and variables to correctly formulate parsimonious mathematical expressions that govern the Lorenz system, outperforming the baseline methods of Eureqa, pySINDy and NGGP.

## 7 EXPERIMENTAL DYNAMICS DISCOVERY: DOUBLE PENDULUM

This section shows SPL-based discovery of a chaotic double pendulum system with experimental data (Asseman et al., 2018) as shown in Appendix Figure E.3. The governing equations are given by:

$$\dot{\omega}_1 = c_1\dot{\omega}_2\cos(\Delta\theta) + c_2\omega_2^2\sin(\Delta\theta) + c_3\sin(\theta_1) + \mathcal{R}_1(\theta_1, \theta_2, \dot{\theta}_1, \dot{\theta}_2),$$
$$\dot{\omega}_2 = c_1\dot{\omega}_1\cos(\Delta\theta) + c_2\omega_1^2\sin(\Delta\theta) + c_3\sin(\theta_2) + \mathcal{R}_2(\theta_1, \theta_2, \dot{\theta}_1, \dot{\theta}_2) \tag{4}$$

where $\theta_1$, $\theta_2$ denote the angular displacements; $\omega_1 = \dot{\theta}_1$, $\omega_2 = \dot{\theta}_2$ the velocities; $\dot{\omega}_1$, $\dot{\omega}_2$ the accelerations; $\mathcal{R}_1$ and $\mathcal{R}_2$ denote the unknown damping forces. Note that $\Delta\theta = \theta_1 - \theta_2$.

The data source contains multiple camera-sensed datasets. Here, 5,000 denoised random sub-samples from 5 datasets are used for training, 2,000 random sub-samples from another 2 datasets for validation, and 1 dataset for testing. The derivatives of the system states are numerically estimated by the same approach discussed in the Lorenz case. Some prior physics knowledge is employed to guide the discovery: **(1)** the terms $\dot{\omega}_2\cos(\Delta\theta)$ for $\dot{\omega}_1$ and $\dot{\omega}_1\cos(\Delta\theta)$ for $\dot{\omega}_2$ are assumed to be part of the governing equations based on the Lagrange derivation; **(2)** the angles ($\theta_1$, $\theta_2$, $\Delta\theta$) are under the trigonometric functions $\cos(\cdot)$ and $\sin(\cdot)$; **(3)** directions of velocities/relative velocities may appear in damping. Production rules fulfilling the above prior knowledge are exhibited in Appendix Section E.2. The hyperparameters are set as $\eta = 1$, $t_{max} = 20$, and 40,000 episodes of training are regarded as one trail. 5 independent trials are performed and the equations with the highest validation scores are selected as the final results. The uncovered equations are given as follows:

$$\dot{\omega}_1 = -0.0991\dot{\omega}_2\cos(\Delta\theta) - 0.103\omega_2^2\sin(\Delta\theta) - 69.274\sin(\theta_1) + \underline{0.515\cos(\theta_1)},$$
$$\dot{\omega}_2 = -1.368\dot{\omega}_1\cos(\Delta\theta) + 1.363\omega_1^2\sin(\Delta\theta) - 92.913\sin(\theta_2) + \underline{0.032\omega_1}, \tag{5}$$

where the explicit expression of physics in an ideal double pendulum system, as displayed in Eq. (5), are successfully distilled and damping terms are estimated. This set of equation is validated through interpolation on the testing set and compared with the smoothed derivatives, as shown in Appendix Figure E.4. The solution appears felicitous as the governing equations of the testing responses.

## 8 CONCLUSION AND DISCUSSION

This paper introduces a Symbolic Physics Learner (SPL) machine to tackle the challenge of distilling the mathematical structure of equations for physical systems (e.g., nonlinear dynamics) with scarce/noisy data. This framework is built upon the expression tree interpretation of mathematical operations and variables and an MCTS agent that searches for the optimistic policy to reconstruct the target mathematical formula. With some remarkable adjustments to the MCTS algorithms, the SPL model straightforwardly accepts our prior or domain knowledge, or any sort of constraints of the tasks in the grammar design while leveraging great flexibility in expression formulation. The robustness of the proposed SPL machine for complex target expression discovery within a large search space is indicated in the Nguyen's symbolic regression benchmark problems, where the SPL machine outperforms state-of-the-art symbolic regression methods. Moreover, encouraging results are obtained in the tasks of discovering physical laws and nonlinear dynamics, based on synthetic or experimental datasets. While the proposed SPL machine shows huge potential in both symbolic regression and physical law discovery tasks, there are still some imperfections that can be improved: (i) the computational cost is high for constant coefficient value estimation due to repeated calls for an optimization process, (ii) graph modularity is underexamined, and (iii) robustness against extreme data noise and scarcity is not optimal. These limitations are further explained in Appendix Section F.

ACKNOWLEDGMENTS

The work is supported by the National Natural Science Foundation of China (No. 92270118) and the Beijing Outstanding Young Scientist Program (No. BJJWZYJH012019100020098).

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

## APPENDIX

## A    HYPERPARAMETER SETTING

We perform a parametric study on the value of discount factor $\eta$ based on Nguyen's benchmark problems without measurement noise. Empirically, setting $\eta = 0.9999$ ensures the scores of ground truth equations stand out and successfully enforces the sparsity in all experiments. For the discovery of very chaotic dynamical systems based on measurement data, we expect some physics terms from the governing equations to have a weak impact on the state variables (e.g., in the chaotic double pendulum system experiments, the effects from physics terms $\sin(\theta_1)$ and $\sin(\theta_2)$ are hard to be captured), and the effect of data noise is unknown. Hence, we set $\eta = 1$ to leverage the full strength of data fitting to enable the detection of physics terms that are offset or overwhelmed by data noise but are pivotal to the systems. Nevertheless, we must acknowledge that this selection process is empirical, which depends on our desire for the degree of parsimony of the target equation(s).

As for the hyperparameters in the training schema (i.e., maximum module transplantation, episodes, maximum tree size, maximum augmented grammars), we have conducted parametric convergence tests for each experiment to ensure the learning curves (i.e., maximum scores in the history) converge. For example, as discussed in Section B, Table B.2 shows the setting of these hyperparameters.

## B    NGUYEN'S BENCHMARK PROBLEMS

This section provides more detailed experiment settings for the Nguyen's benchmark tasks that are described in Section 4.2 of the main text, where the training hyperparameters for the SPL machine in these equation discovery experiments are also listed. Table B.1 presents the candidate mathematical operations allowed for three tested models and Table B.2 displays training hyperparameters for the SPL machine in the Nguyen's benchmark tasks.

Moreover, the utilization of CFG in the SPL machine facilitates the flexibility of applying some prior knowledge including the universal mathematical rules and constraints. This feature empirically turns out to be an accessible and scalable approach for avoiding meaningless mathematical expressions. In the Nguyen's benchmark tasks, one or multiple mathematical constraints are given to the SPL machine. These constraints include

1. Only variables and constant values are allowed in trigonometric functions.

Table B.1: Candidate operators for each Nguyen's benchmark task. $const$ denotes constant values.

| Benchmark | Candidate Operations |
|-----------|----------------------|
| Nguyen-1 | $+, -, \times, \div, \cos(\cdot), \sin(\cdot), \exp(\cdot)$ |
| Nguyen-2 | $+, -, \times, \div, \cos(\cdot), \sin(\cdot), \exp(\cdot)$ |
| Nguyen-3 | $+, -, \times, \div, \cos(\cdot), \sin(\cdot), \exp(\cdot)$ |
| Nguyen-4 | $+, -, \times, \div, \cos(\cdot), \sin(\cdot), \exp(\cdot)$ |
| Nguyen-5 | $+, -, \times, \div, \cos(\cdot), \sin(\cdot), \exp(\cdot)$ |
| Nguyen-6 | $+, -, \times, \div, \cos(\cdot), \sin(\cdot), \exp(\cdot)$ |
| Nguyen-7 | $+, -, \times, \div, \cos(\cdot), \sin(\cdot), \exp(\cdot), \log(\cdot), \sqrt{\cdot}$ |
| Nguyen-8 | $+, -, \times, \div, \cos(\cdot), \sin(\cdot), \exp(\cdot), \log(\cdot), \sqrt{\cdot}$ |
| Nguyen-9 | $+, -, \times, \div, \cos(\cdot), \sin(\cdot), \exp(\cdot)$ |
| Nguyen-10 | $+, -, \times, \div, \cos(\cdot), \sin(\cdot), \exp(\cdot)$ |
| Nguyen-11 | $+, -, \times, \div, \cos(\cdot), \sin(\cdot), \exp(\cdot), \log(\cdot), \sqrt{\cdot}$ |
| Nguyen-12 | $+, -, \times, \div, \cos(\cdot), \sin(\cdot), \exp(\cdot)$ |
| Nguyen-1$^c$ | $+, -, \times, \div, \cos(\cdot), \sin(\cdot), \exp(\cdot), const$ |
| Nguyen-2$^c$ | $+, -, \times, \div, \cos(\cdot), \sin(\cdot), \exp(\cdot), const$ |
| Nguyen-5$^c$ | $+, -, \times, \div, \cos(\cdot), \sin(\cdot), \exp(\cdot), const$ |
| Nguyen-8$^c$ | $+, -, \times, \div, \cos(\cdot), \sin(\cdot), \exp(\cdot), \log(\cdot), \sqrt{\cdot}, const$ |
| Nguyen-9$^c$ | $+, -, \times, \div, \cos(\cdot), \sin(\cdot), \exp(\cdot), const$ |

Table B.2: Training Hyperparameter settings for the SPL model in Nguyen's benchmark tasks.

| Benchmark | Maximum Module Transplantation | Episodes Between Module Transplantation | Maximum Tree Size | Maximum Augmented Grammars |
|---|---|---|---|---|
| Nguyen-1 | 20 | 10,000 | 50 | 5 |
| Nguyen-2 | 20 | 10,000 | 50 | 5 |
| Nguyen-3 | 20 | 100,000 | 50 | 5 |
| Nguyen-4 | 20 | 100,000 | 50 | 5 |
| Nguyen-5 | 20 | 100,000 | 50 | 5 |
| Nguyen-6 | 20 | 10,000 | 50 | 5 |
| Nguyen-7 | 20 | 5,000 | 50 | 5 |
| Nguyen-8 | 20 | 5,000 | 50 | 5 |
| Nguyen-9 | 20 | 10,000 | 50 | 5 |
| Nguyen-10 | 20 | 10,000 | 50 | 5 |
| Nguyen-11 | 20 | 10,000 | 50 | 5 |
| Nguyen-12 | 20 | 100,000 | 50 | 5 |
| Nguyen-1$^c$ | 20 | 2,000 | 50 | 5 |
| Nguyen-2$^c$ | 20 | 10,000 | 50 | 5 |
| Nguyen-5$^c$ | 20 | 10,000 | 50 | 5 |
| Nguyen-8$^c$ | 20 | 2,000 | 50 | 5 |
| Nguyen-9$^c$ | 20 | 1,000 | 50 | 5 |

Table B.3: Mathematical constraints for the SPL machine in each Nguyen's benchmark problem.

| Benchmark | Constraints Utilization |
|---|---|
| Nguyen-1 | {1, 3} |
| Nguyen-2 | {1, 3} |
| Nguyen-3 | {1, 3} |
| Nguyen-4 | {1, 3} |
| Nguyen-5 | {2, 3} |
| Nguyen-6 | {2, 3} |
| Nguyen-7 | {2, 3} |
| Nguyen-8 | {} |
| Nguyen-9 | {2, 3, 4} |
| Nguyen-10 | {2, 3, 4} |
| Nguyen-11 | {2, 3} |
| Nguyen-12 | {1, 3, 4} |
| Nguyen-1$^c$ | {1, 3} |
| Nguyen-2$^c$ | {1, 3} |
| Nguyen-5$^c$ | {2, 3} |
| Nguyen-8$^c$ | {} |
| Nguyen-9$^c$ | {2, 3, 4} |

2. Variables in trigonometric functions, logarithms and roots are up to the polynomial of 3.

3. Trigonometric functions, logarithms and roots are not allowed to form unreasonable composite functions with each other, such as $\sin(\cos(...))$.

4. Some small integers (e.g. 1, 2) are used directly as leaves.

They can be easily implemented into the SPL machine by defining or adjusting non-terminal nodes or production rules in the customized CFG. Constraints adopted by each task are shown in Table B.3.

## C    RESULTS OF ABLATION STUDY

We consider four ablation studies by removing the following:

    (a) the adaptive scaling in reward calculation,

    (b) the discount factor $\eta^n$ that drives equation parsimony in Eq. (2),

    (c) module transplantation in tree generation,

    (d) all of the above three.

The resulting models are denoted with Model A, Model B, Model C, and Model D (note that Model D is equivalent to the vanilla MCTS). The ablation study is performed on the 12 classic Nguyen's benchmark problems, where the recovery rate of each model is calculated. The results of the ablation study are summarized in Table C.1. It is clear that module transplantation brings the largest gain in recovery rate. All the ablated models fail to uncover the Nguyen-12 equation (the most difficult case), where the coefficient 1/2 needs to be represented by mathematical operators and symbols, e.g., $x/(x + x)$, $y/(y + y)$, etc.

## D    FREE FALLING BALLS WITH AIR RESISTANCE

This appendix section reveals more details on discovering the physical laws, in the context of relationships between height and time, for the cases of free-falling objects with air resistance based on multiple experimental ball-drop datasets (de Silva et al., 2020). The datasets contain the records of 11 different types of balls, as shown in Figure D.1, dropped from a bridge, collected at a 30 Hz sampling rate. The time between dropping and landing varies in each case due to the fact that air resistance has different effects on these free-dropping balls and induces divergent physical laws. Consequently, we consider each ball as an individual experiment and discover the physical law for each of them. The measurement dataset of a free-dropping ball is split into a training set (records from the first 2 seconds, 60 measurements) and a testing set (records after 2 seconds).

The physical laws of the three baseline models and distilled by the SPL machine from training data are exhibited in Table D.1. These discovered physical laws are then applied to forecast the height of the balls at the time slots in the testing dataset. These predictions, in comparison with the ground truth trajectory recorded, are shown in Figure D.2. The prediction error is shown in the Table 3 of the main text.

## E    NONLINEAR DYNAMICS DISCOVERY

This appendix section elaborates more details on the two governing equation discovery experiments presented in the main text, including the datasets and full results.

Table C.1: Summary of the ablation study results.

| Benchmark | Expression | SPL | Model A | Model B | Model C | Model D |
|---|---|---|---|---|---|---|
| Nguyen-1 | $x^3 + x^2 + x$ | 100% | 100% | 100% | 14% | 12% |
| Nguyen-2 | $x^4 + x^3 + x^2 + x$ | 100% | 100% | 100% | 0% | 0% |
| Nguyen-3 | $x^5 + x^4 + x^3 + x^2 + x$ | 100% | 100% | 100% | 0% | 0% |
| Nguyen-4 | $x^6 + x^5 + x^4 + x^3 + x^2 + x$ | 99% | 96% | 92% | 0% | 0% |
| Nguyen-5 | $\sin(x^2)\cos(x) - 1$ | 95% | 95% | 92% | 92% | 88% |
| Nguyen-6 | $\sin(x^2) + \sin(x + x^2)$ | 100% | 100% | 96% | 100% | 100% |
| Nguyen-7 | $\ln(x + 1) + \ln(x^2 + 1)$ | 100% | 100% | 100% | 100% | 100% |
| Nguyen-8 | $\sqrt{x}$ | 100% | 100% | 100% | 100% | 100% |
| Nguyen-9 | $\sin(x) + \sin(y^2)$ | 100% | 100% | 100% | 100% | 100% |
| Nguyen-10 | $2\sin(x)\cos(y)$ | 100% | 100% | 100% | 0% | 0% |
| Nguyen-11 | $x^y$ | 100% | 96% | 96% | 92% | 87% |
| Nguyen-12 | $x^4 - x^3 + \frac{1}{2}y^2 - y$ | 28% | 0% | 0% | 0% | 0% |
| Average | | **93.5%** | 90.58% | 89.67% | 49.83% | 48.92% |

Table D.1: Uncovered physics from the motions of free-falling balls by the SPL machine and three baseline models. Note that these formulas are the raw equations produced by SRL. Further simplification helps better parsimony of the formulas.

| Type | Model | Expression |
|---|---|---|
| baseball | SPL | $H(t) = 47.8042 + 0.6253t - 4.5383t^2$ |
| | Model-1 | $H(t) = 47.682 + 1.456t - 5.629t^2 + 0.376t^3$ |
| | Model-2 | $H(t) = 45.089 - 8.156t + 5.448 \exp(0t)$ |
| | Model-3 | $H(t) = 48.051 - 183.467 \log(\cosh(0.217t))$ |
| blue basketball | SPL | $H(t) = 46.4726 - 5.105t^2 + t^3 - 0.251t^4$ |
| | Model-1 | $H(t) = 46.513 - 0.493t - 3.912t^2 + 0.03t^3$ |
| | Model-2 | $H(t) = 43.522 - 7.963t + 5.306 \exp(0t)$ |
| | Model-3 | $H(t) = 46.402 - 84.791 \log(\cosh(0.319t))$ |
| green basketball | SPL | $H(t) = 45.9087 - 4.1465t^2 + \log(\cosh(1))$ |
| | Model-1 | $H(t) = 46.438 - 0.34t - 3.882t^2 - 0.055t^3$ |
| | Model-2 | $H(t) = 43.512 - 8.043t + 5.346 \exp(0t)$ |
| | Model-3 | $H(t) = 46.391 - 124.424 \log(\cosh(0.263t))$ |
| volleyball | SPL | $H(t) = 48.0744 - 3.7772t^2$ |
| | Model-1 | $H(t) = 48.046 + 0.362t - 4.352t^2 + 0.218t^3$ |
| | Model-2 | $H(t) = 45.32 - 7.317t + 5.037 \exp(0t)$ |
| | Model-3 | $H(t) = 48.124 - 107.816 \log(\cosh(0.27t))$ |
| bowling ball | SPL | $H(t) = 46.1329 - 3.8173t^2 - 0.2846t^3 + 4.14 \times 10^{-5} \exp(20.7385t^2) \exp(-12.4538t^3)$ |
| | Model-1 | $H(t) = 46.139 - 0.091t - 3.504t^2 - 0.431t^3$ |
| | Model-2 | $H(t) = 43.336 - 8.525t + 5.676 \exp(0t)$ |
| | Model-3 | $H(t) = 46.342 - 247.571 \log(\cosh(0.189t))$ |
| golf ball | SPL | $H(t) = 49.5087 - 4.9633t^2 + \log(\cosh(t))$ |
| | Model-1 | $H(t) = 49.413 + 0.532t - 5.061t^2 + 0.102t^3$ |
| | Model-2 | $H(t) = 46.356 - 8.918t + 5.964 \exp(0t)$ |
| | Model-3 | $H(t) = 49.585 - 178.47 \log(\cosh(0.23t))$ |
| tennis ball | SPL | $H(t) = 47.8577 - 4.0574t^2 + \log(\cosh(0.121t^3))$ |
| | Model-1 | $H(t) = 47.738 + 0.658t - 4.901t^2 + 0.325t^3$ |
| | Model-2 | $H(t) = 45.016 - 7.717t + 5.212 \exp(0t)$ |
| | Model-3 | $H(t) = 47.874 - 114.19 \log(\cosh(0.269t))$ |
| whiffle ball 1 | SPL | $H(t) = 4.1563t^2 - t^3 + 47.0133 \exp(-0.1511t^2)$ |
| | Model-1 | $H(t) = 46.969 + 0.574t - 4.505t^2 + 0.522t^3$ |
| | Model-2 | $H(t) = 44.259 - 6.373t + 4.689 \exp(0t)$ |
| | Model-3 | $H(t) = 47.062 - 34.083 \log(\cosh(0.462t))$ |
| whiffle ball 2 | SPL | $H(t) = -18.6063 + 65.8583 \exp(-0.0577t^2)$ |
| | Model-1 | $H(t) = 47.215 + 0.296t - 4.379t^2 + 0.421t^3$ |
| | Model-2 | $H(t) = 44.443 - 6.744t + 4.813 \exp(0t)$ |
| | Model-3 | $H(t) = 47.255 - 38.29 \log(\cosh(0.447t))$ |
| yellow whiffle ball | SPL | $H(t) = 148.9911/(\log(\cosh(t)) + 3.065) - 14.5828t^2/(\log(\cosh(t)) + 3.065)$ $+ 48.6092 \log(\cosh(t))/(\log(\cosh(t)) + 3.065)$ |
| | Model-1 | $H(t) = 48.613 - 0.047t - 4.936t^2 + 0.826t^3$ |
| | Model-2 | $H(t) = 45.443 - 6.789t + 4.973 \exp(0t)$ |
| | Model-3 | $H(t) = 48.594 - 12.49 \log(\cosh(0.86t))$ |
| orange whiffle ball | SPL | $H(t) = -1.6626t + 47.8622 \exp(-0.06815t^2)$ |
| | Model-1 | $H(t) = 47.836 - 1.397t - 3.822t^2 + 0.422t^3$ |
| | Model-2 | $H(t) = 44.389 - 7.358t + 5.152 \exp(0t)$ |
| | Model-3 | $H(t) = 47.577 - 12.711 \log(\cosh(0.895t))$ |

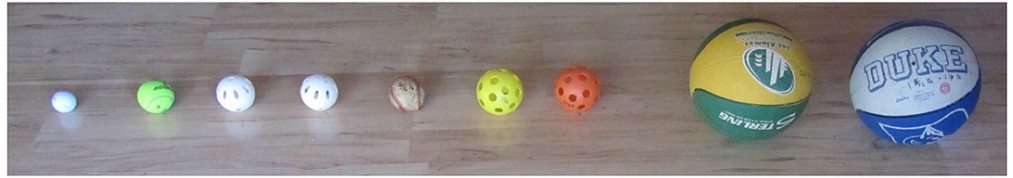

Figure D.1: The experimental balls that were dropped from the bridge (de Silva et al., 2020). From left to right: golf ball, tennis ball, whiffle ball 1, whiffle ball 2, baseball, yellow whiffle ball, orange whiffle ball, green basketball, and blue basketball. Volleyball is not shown here.

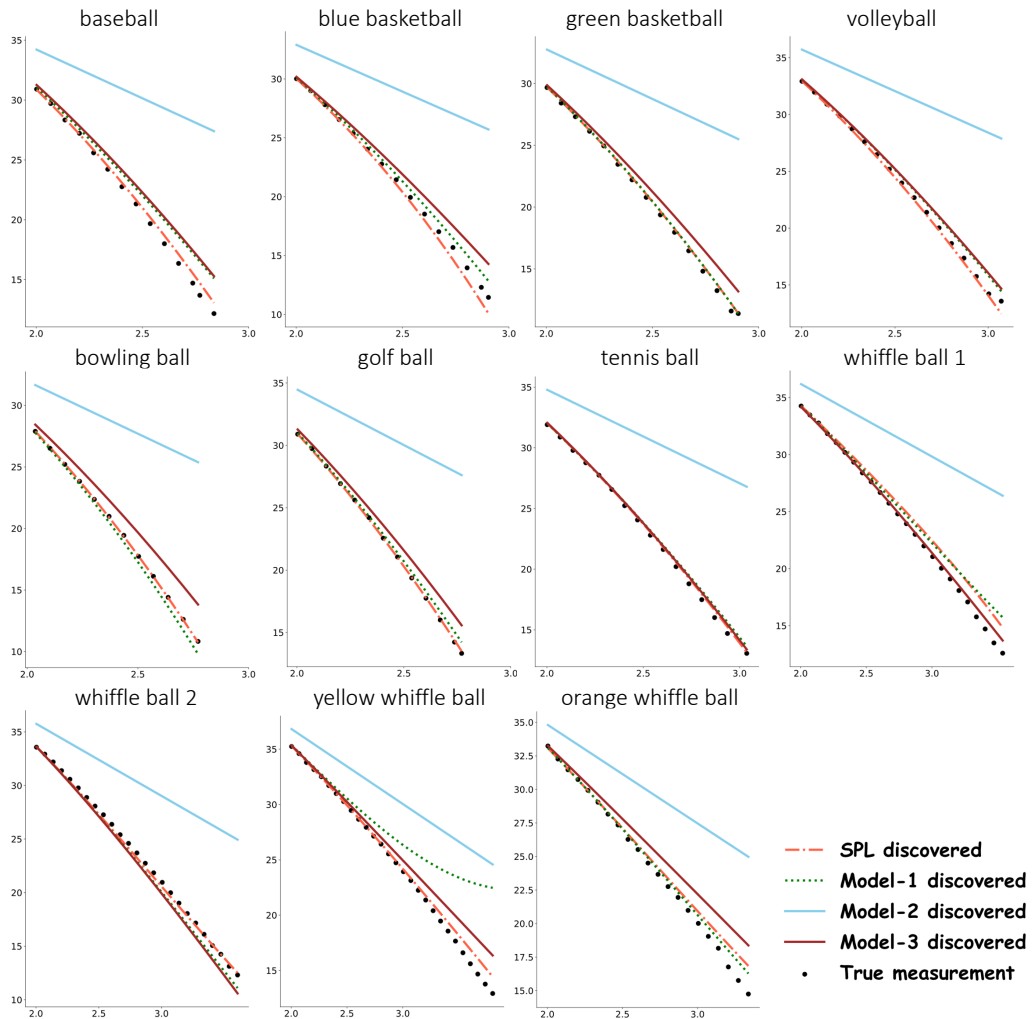

Figure D.2: Trajectories after 2 seconds predicted by uncovered physical laws.

### E.1 LORENZ SYSTEM

The 3-dimensional Lorenz system is governed by

$$\begin{aligned}
\dot{x} &= \sigma(y - x) \\
\dot{y} &= x(\rho - z) - y \\
\dot{z} &= xy - \beta z
\end{aligned} \tag{6}$$

with parameters $\sigma = 10$, $\beta = 8/3$, and $\rho = 28$, under which the Lorenz attractor has two lobes and the system, starting from anywhere, makes cycles around one lobe before switching to the

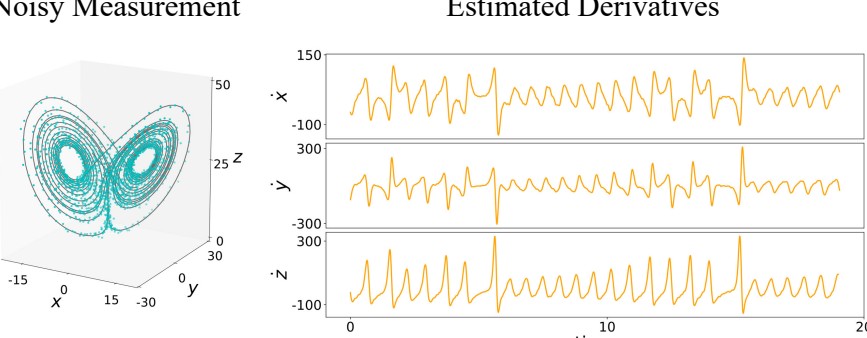

Figure E.1: Lorenz system for the experiment. Noisy measurement data and numerically estimated derivatives smoothed by Savitzky–Golay filter.

Table E.1: Discovered governing equations for the Lorenz system.

| Model | Discovered Governing Equations |
|---|---|
| Eureqa | $\dot{x} = -0.56 - 9.02x + 9.01y$
$\dot{y} = -0.047 + 18.79x + 1.86y - 0.046xy - 0.74xz$
$\dot{z} = -3.04 - 2.23z + 0.88xy$ |
| pySINDy | $\dot{x} = -0.46 - 9.18x + 9.17y$
$\dot{y} = 22.32x + 0.15y - 0.85xz$
$\dot{z} = 6.04 - 2.83z + 0.15x^2 + 0.81xy$ |
| NGGP | $\dot{x} = 0.0047 - 10.02x + 10.01y - 0.007x^2 + 0.007xy - 0.37x/z - 0.00074x^2y$
$\quad + 0.00063x^3 + 0.00018x^2z + 0.00011xy^2 - 6.59e^{-5}xyz - 0.00011y^z$
$\dot{y} = 26.36x - 1.5y - 0.83xz - 7.20x/z + 13.08y/z + 4.52e^{-5}x^3 - 0.0038xz^2$
$\quad - 44.25x/z^2 + 0.00028x^3/z - 0.00017x^3z + 5.98e^{-6}x^3z^2$
$\dot{z} = -0.64 - 0.036y - 2.64z + 1.038xy + 0.00021xz + 0.0011yz - 0.00021x/z$
$\quad - 8.04e^{-8}y/x + 0.00022y/z - 8.04e^{-8}z^2 - 0.00021xyz + 0.00021xy/z$
$\quad - 0.001y^2z - 0.00021y^2/z + 1.17y/z^2 - 3.89e^{-7}yz^2/x + 6.66e^{-9}z^2/x$ |
| SPL | $\dot{x} = -9.966x + 9.964y$
$\dot{y} = 27.764x - 0.942y - 0.994xz$
$\dot{z} = -2.655z + 0.996xy$ |
| True | $\dot{x} = -10x + 10y$
$\dot{y} = 28x - y - xz$
$\dot{z} = -2.667z + xy$ |

other and iterates repeatedly. The measurement data of the Lorenz system states in this experiment contains a clean signal with 5% Gaussian white noise. The derivatives of Lorenz's state variables are unmeasured but numerically estimated and smoothed by the Savitzky–Golay filter. The noisy synthetic measurement data and numerically obtained derivatives are shown in Figure E.1.

Table E.1 presents the distilled governing equations for the Lorenz system by the SPL machine compared with 3 baseline models. It is observed that the SPL machine uncovers the explicit form of equations accurately in the context of active terms, whereas Eureqa, pySINDy and NGGP yield several false-positive terms in the underlying governing equations. The predicted system responses (starting from a different initial condition) simulated from these uncovered equations are shown in Figure E.2. Although it is hard to reproduce the most accurate coefficients due to the tremendous errors induced by numerical differentiation of noisy measurement data as depicted in Figure E.1, the SPL machine is still capable of distilling the most concise symbolic combination of operators and variables to correctly formulate the parsimonious mathematical expressions that govern the Lorenz system dynamics. The predicted responses for the governing equations unearthed by the SPL machine simulate the system in a decent manner.

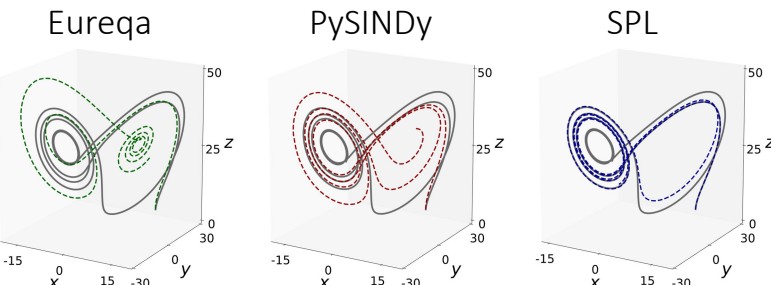

Figure E.2: Response prediction for 5 seconds by identified governing equations (dashed plots) under a different validation IC of Lorenz system, in comparison with the ground truth trajectory (grey).

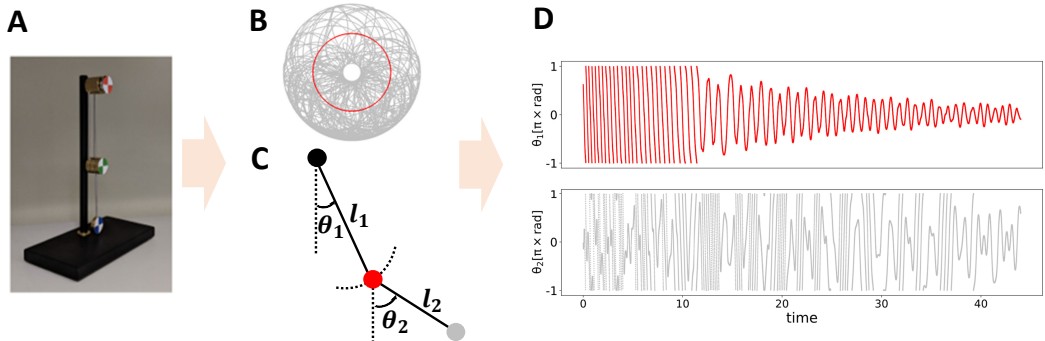

Figure E.3: Double Pendulum system experiment and measurement data (Asseman et al., 2018): A. experiment setup. B. displacements of the two moving masses. C. model the system with $\theta_1$ and $\theta_2$. D. angles of two masses transformed from displacements.

## E.2 MOUNTED DOUBLE PENDULUM SYSTEM

The second nonlinear dynamics discovery experiment is a chaotic double pendulum system (Asseman et al., 2018). The measured data, in form of videos, represents the chaotic motion of a double pendulum on the device shown in Figure E.3A filmed with a high-speed camera. The positional data is converted into angular form based on the geometry information (see the model shown in Figure E.3C). The governing equations can be derived using the Euler–Lagrange method, given by

$$(m_1 + m_2)l_1\ddot{\theta}_1 + m_2l_2\ddot{\theta}_2\cos(\theta_1 - \theta_2) + m_2l_2\omega_2^2\sin(\theta_1 - \theta_2) + (m_1 + m_2)g\sin(\theta_1) = F_1,$$
$$m_2l_2\ddot{\theta}_2 + m_2l_1\ddot{\theta}_1\cos(\theta_1 - \theta_2) - m_2l_1\omega_1^2\sin(\theta_1 - \theta_2) + m_2g\sin(\theta_2) = F_2,$$

$$(7)$$

which, by denoting $\omega$ as the velocity, can be converted to the following state-space form:

$$\dot{\theta}_1 = \omega_1,$$
$$\dot{\theta}_2 = \omega_2,$$
$$\dot{\omega}_1 = c_1\dot{\omega}_2\cos(\Delta\theta) + c_2\omega_2^2\sin(\Delta\theta) + c_3\sin(\theta_1) + \mathcal{R}_1(\theta_1, \theta_2, \dot{\theta}_1, \dot{\theta}_2),$$
$$\dot{\omega}_2 = c_1\dot{\omega}_1\cos(\Delta\theta) + c_2\omega_1^2\sin(\Delta\theta) + c_3\sin(\theta_2) + \mathcal{R}_2(\theta_1, \theta_2, \dot{\theta}_1, \dot{\theta}_2)$$

$$(8)$$

where $\Delta\theta = \theta_1 - \theta_2$, and $\mathcal{R}_1(\theta_1, \theta_2, \dot{\theta}_1, \dot{\theta}_2)$ and $\mathcal{R}_2(\theta_1, \theta_2, \dot{\theta}_1, \dot{\theta}_2)$ denote the damping terms for the last two differential equations.

The data source contains multiple video datasets. For this discovery, 5,000 denoised random sub-samples from 5 datasets are used for training purposes, and 2,000 random sub-samples from another 2 datasets for validation, and 1 dataset for testing. Some prior knowledge guiding this discovery includes:

Table E.2: Discovered governing equations of the mounted double pendulum by the SPL model.

| Phase | Expression |
|---|---|
| $\dot{\omega}_1$ | $-0.0991\dot{\omega}_2\cos(\Delta\theta) - 0.103\omega_2^2\sin(\Delta\theta) - 69.274\sin(\theta_1) + 0.515\cos(\theta_1)$ |
| $\dot{\omega}_2$ | $-1.368\dot{\omega}_1\cos(\Delta\theta) + 1.363\omega_1^2\sin(\Delta\theta) - 92.913\sin(\theta_2) + 0.032\omega_1$ |

1. the two terms $\dot{\omega}_2\cos(\Delta\theta)$ for $\dot{\omega}_1$ and $\dot{\omega}_1\cos(\Delta\theta)$ for $\dot{\omega}_2$, which can be easily derived based on our prior knowledge on the system, are assumed known in the two governing equations. However, their coefficients are unknown and need to be estimated.

2. the remaining of the formulas are potentially comprised of the free combination of $\dot{\omega}_1$, $\dot{\omega}_2$, $\omega_1$, $\omega_2$, as well as the angles ($\theta_1$, $\theta_2$, $\Delta\theta$) under the trigonometric functions $\cos(\cdot)$ and $\sin(\cdot)$.

3. velocities and relative velocities of two masses, as well as their directions (sign function) might contribute to the damping.

Based on the above information, the candidate production rules for $\dot{\omega}_1$ and $\dot{\omega}_2$ equations are shown below, where non-terminal nodes are $V = \{A, W, T, S\}$ and $C$ denotes the placeholder symbol for the constant coefficient values.

$$A \to A + A, A \to A \times A, A \to C, A \to A + A, A \to W,$$
$$W \to W \times W, W \to \omega_1, W \to \omega_2, W \to \dot{\omega}_1, W \to \dot{\omega}_2,$$
$$A \to \cos(T), A \to \sin(T), T \to T + T, T \to T - T, T \to \theta_1, T \to \theta_2,$$
$$A \to sign(S), S \to S + S, S \to S - S, S \to \omega_1, S \to \omega_2, S \to \dot{\omega}_1, S \to \dot{\omega}_2,$$
$$A \to \dot{\omega}_1\cos(\theta_1 - \theta_2), A \to \dot{\omega}_2\cos(\theta_1 - \theta_2).$$

Note that our prior knowledge can be easily incorporated in the proposed SPL machine to improve the discovery performance, rather than relying on the free combination of mathematical operators and symbols. The hyperparameters are set as $\eta = 1$, $t_{max} = 20$, and 40,000 episodes of training are regarded as one trail. 5 independent trials are performed and the equations with the highest validation scores are selected as the final result. The uncovered equations are shown in Table E.2. They are validated through interpolation on the testing set and compared with the smoothed derivatives, as shown in Figure E.4. The solution appears felicitous as the governing equations of the testing responses.

## F    DISCUSSION AND FUTURE DIRECTIONS

The effectiveness of the proposed SPL machine is empowered by the following elements: (1) The use of MCTS enables the flexible representation of search space with customized computational grammars, composed of a finite set of mathematical operators and symbols, to guide the search tree expansion. (2) The exploration-exploitation trade-off nature of MCTS is remarkably useful for searching the optimal mathematical expression tree. (3) The key adjustments, including the greedy search, the adaptive-scaled rewarding, the reward regularizer, and the expression tree module transplantation, make it possible to efficiently uncover the best path to formulate complex equations. (4) The SPL machine straightforwardly accepts our prior or domain knowledge, or any sort of constraints of the tasks in the grammar design while leveraging great flexibility in expression formulation.

While SPL shows huge potential in both symbolic regression and governing equation discovery tasks, there are still some imperfections to be improved. In this appendix section, a few bottlenecks and their potential solutions are presented:

1. **Computational cost.** Computational cost for this framework is one of the major issues, especially when constant coefficient estimation is required. Evaluating the solution in the simulation phase happens very frequently for the MCTS algorithm where the policy selection relies heavily on a large number of historical rewards. However, the constant coefficient value estimation, which requires repeated calls for an optimization process and can be slow,

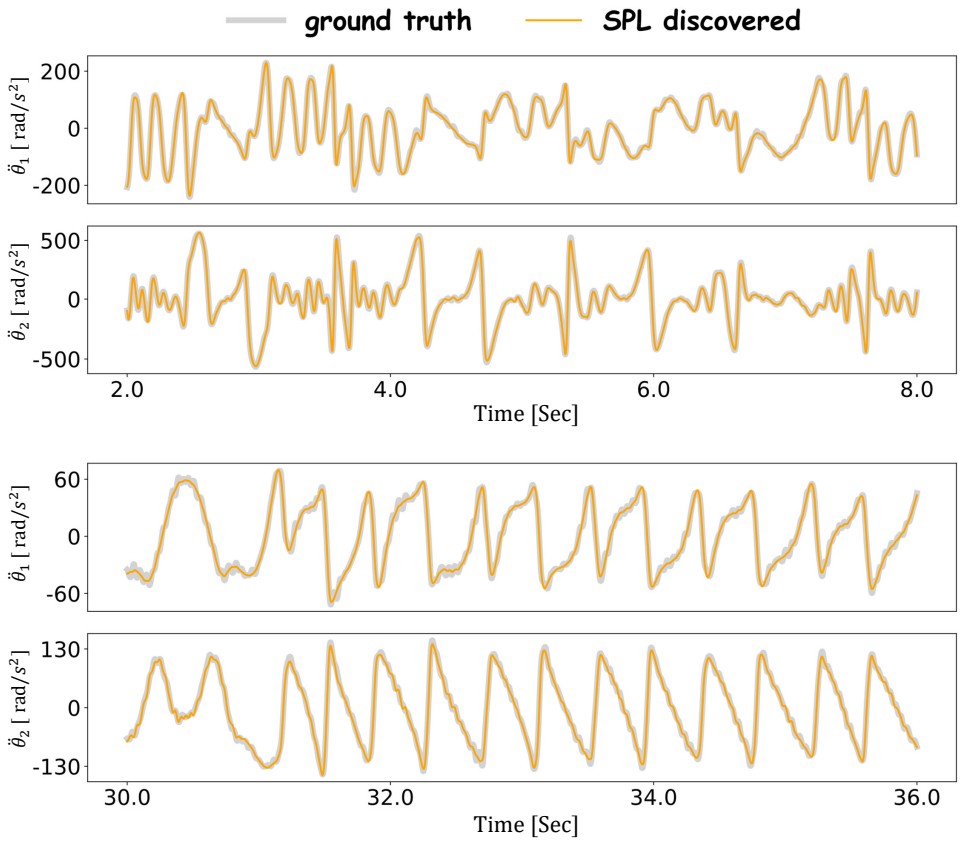

Figure E.4: Discovered governing equations of the mounted double pendulum system on a different dataset in different time sections: in 2-8 seconds the two masses are in chaotic motions while in 30-36 seconds the masses tend to move periodically due to accumulative damping. $\ddot{\theta}_1$ and $\ddot{\theta}_2$ are obtained through smoothed numerical differentiation and predicted from the discovered governing equations.

Table E.3: Average training time (in seconds) of SPL and NGGP in the Nguyen's benchmark problems

| Benchmark | SPL [s] | NGGP [s] |
|---|---|---|
| Nguyen-1 | 8.776 | 2.734 |
| Nguyen-2 | 7.296 | 3.296 |
| Nguyen-3 | 81.287 | 3.945 |
| Nguyen-4 | 567.061 | 5.764 |
| Nguyen-5 | 431.228 | 77.627 |
| Nguyen-6 | 64.651 | 104.588 |
| Nguyen-7 | 14.995 | 3.024 |
| Nguyen-8 | 5.59 | 2.896 |
| Nguyen-9 | 5.743 | 13.229 |
| Nguyen-10 | 53.245 | 86.497 |
| Nguyen-11 | 10.163 | 44.399 |
| Nguyen-12 | 187.9 | 334.757 |
| Nguyen-1$^c$ | 452.734 | 362.075 |
| Nguyen-2$^c$ | 295.769 | 1188.215 |
| Nguyen-5$^c$ | 2178.891 | 1365.777 |
| Nguyen-8$^c$ | 77.892 | 129.349 |
| Nguyen-9$^c$ | 2001.402 | 3066.41 |

is needed for evaluation purposes. In particular, the constant coefficient value is estimated via concurrently solving an optimization problem, e.g., by Powell's conjugate direction method (Powell, 1964). For example, when the tree structure changes or is updated, the optimization of the constant coefficients should be re-performed simultaneously. The SPL machine is not the only symbolic regressor suffering from the computational cost in constant coefficient value estimation. In fact, the state-of-the-art symbolic regression model, the neural-guided GP (NGGP) (Mundhenk et al., 2021), becomes much slower in the Nguyen's benchmark variant tasks (see Table E.3). The current implementation of the SPL machine tries to empirically avoid this issue by limiting the number of placeholders in a discovered expression and simplifying the expression before evaluation, but still cannot reach great efficiency. This bottleneck might be mitigated if parallel computing is introduced to the MCTS simulation phase.

2. **Graph modularity underexamined.** The current design of the SPL training scheme does not leverage the full graph modularity: modules are reached by transforming a complete parse tree into a grammar. However, in some cases, there might be some influential modules appearing frequently as part of the tree. This type of graph modularity is described in the AI-Feynman method (Udrescu et al., 2020). Deploying a more comprehensive graph modularity into the SPL machine will boost its efficacy in the complex equation and nonlinear dynamics discovery tasks.

3. **Robustness against extreme data noise and scarcity.** Although it is observed that this reinforcement learning-based method is able to unearth the parsimonious solution to the governing equations from synthetic or measurement data with a moderate level of noise and scarcity. It is not effective when the data condition is extreme, or if there are missing values that make it challenging to numerically calculate the state derivatives. It is reasonable to investigate the integration between the SPL framework with a differentiable surrogate model built upon neural networks (Long et al., 2018; Chen et al., 2021) or spline learning (Sun et al., 2021) for further robustness in nonlinear dynamics discovery tasks.

