# OpenReview forum: "Symbolic Physics Learner: Discovering governing equations via Monte Carlo tree search"
_ICLR.cc/2023/Conference — ICLR 2023 notable top 5%_

### Official Review · Reviewer_uNr2 · 2022-10-24

**Confidence:** 4
**Correctness:** 3
**Technical Novelty And Significance:** 3
**Empirical Novelty And Significance:** 3
**Recommendation:** 8

**Clarity, Quality, Novelty And Reproducibility:**

There are some missing issues in the paper.

References:

--Need a reference for eq. (1), or a note as to why this is a good measure.

--Sec. (3): the claim "It has been demonstrated that the MCTS agent continuously gains knowledge" needs a reference.

Need a more precise definition of  Recovery rate (Table 1). It is well understood that the recovered equations typically contain many more terms than the target equations, and this needs to be shown in the definition and results. Do you mean "regeneration of the precise equation with no additional terms"?

A key missing factor in the paper (Algorithm 1) is the specification of the regularization terms. Symbolic regression has major problems with limiting the size of the learned equations without good regularization terms; this is not addressed in the body of the paper and must be.





**Strength And Weaknesses:**

Strengths:

This is a strong paper, and has clear novelty in the area of symbolic regression.
The paper has extensive comparison to competing methods, with good results.


Weaknesses:

The theoretical aspects of the paper could be improved.

The theoretical basis of this approach should be highlighted in comparison to existing methods of symbolic regression. In addition, since it is based on stochastic methods for which guarantees have been given for other applications, this is a clear lack. Questions include:
What type of sampling distributions are used, and can these be modified during search to improve the returned function?


**Summary Of The Paper:**

The authors propose a Monte Carlo tree search (MCTS) algorithm to generate optimal expression trees based on measurement data. The method is validated using Nguyen’s symbolic regression benchmark task (Uy et al., 2011), along with several other benchmarks.





**Summary Of The Review:**

This paper presents a novel approach to symbolic regression, and shows good improvement over existing methods. Overall, a strong contribution.

---

> ### Author Response · Authors · 2022-11-17
> **Response to Reviewer uNr2**
>
> We sincerely thank the reviewer for the positive and constructive comments/suggestions, which are very helpful for improving our paper. Please find our responses below. Revisions have also been made in the paper.
>
> **1. Theoretical Aspects:** Thanks for this comment. Since the search engine we employed in our proposed SPL machine is the UCT-based MCTS, the theoretical analysis (e.g.,  convergence, guarantees) of this algorithm is found in literature such as Shah et al. (2019). We have clarified this in our revised manuscript (see Section 2).
>
> **2. Type of Sampling Distributions:** Thanks for this question. We used the most standard uniform distribution to generate random samples. We acknowledge that a more advanced sampling strategy (e.g., sparse sampling) might be helpful to improve the MCTS performance (e.g., efficiency). The reviewer points out an important direction for our future study.
>
> **3. References:** Thanks for your careful reading. The references have been incorporated in our revised manuscript.
>
> **4. Definition of Recovery Rate:** The reviewer is correct. The discovered equation that perfectly fits the testing data is regarded as a successful discovery (i.e., the discovered equation should be identical or equivalent to the target one). We have clarified this in Section 4.2 in the revised manuscript.
>
> **5. Specification of the Regularization Terms:** Thanks for this excellent suggestion. In the proposed SPL machine, the term $\eta^n$ in Eq. 2 denotes the regularizer that controls the equation parsimony, where $\eta$ is the discount factor ($0<\eta<1$). We have clarified this in Algorithm 1.
>
> We hope this response would help clarify the reviewer's concern. Please do not hesitate to let us know if you have any further questions. Thank you very much.

---

### Official Review · Reviewer_X2Dz · 2022-10-24

**Confidence:** 3
**Correctness:** 4
**Technical Novelty And Significance:** 3
**Empirical Novelty And Significance:** 3
**Recommendation:** 8

**Clarity, Quality, Novelty And Reproducibility:**

It is unclear to me how you can predict continuous constant values. Can you elaborate on that?
You mention in the limitations and in the appendix that the computational cost
for constant value estimation is high. Why is that the case?

In table B.1, the learned physics for the yellow whiffle ball and the bowling ball
involve big expressions with large constant values and multiple components.
They seem to be non-parsimonious according to your descriptions. Do these
equations show signs of overfitting, and if so, are there ways to address that in
your method?

In the Lorenz system in sec. 6 and in the double pendulum in sec. 7, what is the importance of Savitzky-Golay filter?
Would your method work without it, or is it integral to compute noise-free
derivatives? Is this filtering approach applicable in a
variety of settings? Could the introduced smoothness be harmful in a
setting without noise?

In the double pendulum in sec. 8, you employ some physics to guide the discovery. Why is that?
Is the overall system too hard to model otherwise?
What is the performance of other methods in this setting?

### Additional

Table 4 is out of the document margins. Please respect the document margins for
the camera ready version.

### Typos

- Abstract: optimistic policy -> optimal policy
- Abstract: PSL -> SPL
- p. 3, sec. 2: stikcs
- p. 3 sec. 3: at an an expandable node
- p. 5, Adaptive-scaled rewarding: unpredicTable


**Strength And Weaknesses:**

The paper is well written and easy to follow.
The authors perform a multitude of experiments on symbolic regression and
discovery of nonlinear dynamics.
Comparison with state-of-the-art related works clearly demonstrates the
shortcomings of related works and the efficacy of the proposed method.

I am missing a more detailed description and direct comparison with related works on MCTS for symbolic regression.
How is your work different from theirs?
Similarly, I would expect a quantitative comparison with existing MCTS methods for symbolic regression.

There is one ablation study on the importance of adjustments to MCTS (sec. 4.2),
yet the results are barely mentioned, and I miss some analysis on them.
You could have included the results (and the analysis) in the appendix.
It is surprising that removing each adjustment separately results in a performance lower than 60%.
Can you provide more detailed results, and perhaps explain why that might be the case?


**Summary Of The Paper:**

This paper proposes Symbolic Physics Learner (SPL), a method for symbolic regression and discovery of dynamics.
The authors formalize mathematical operations using expression trees and context-free grammars,
and employ Monte Carlo Tree Search (MCTS) to explore the space and select
optimal solutions.
Furthermore, they propose adjustments to conventional MCTS, including adaptive
scaling and module transplantation.
The authors demonstrate the efficacy of the proposed method in a number of
settings and achieve superior performance compared to other
state-of-the-art methods.


**Summary Of The Review:**

I recommend that this paper is accepted. It is a well written paper with important technical contributions and strong quantitative results. A comparison with related works on MCTS for symbolic regression could help the readers appreciate its contributions more. More thorough ablation studies on the introduced components could further clarify their efficacy.

---

> ### Author Response · Authors · 2022-11-17
> **Response to Reviewer X2Dz (Part 2)**
>
> **6. Prior Physics in the Double Pendulum Example:** Thanks for this excellent question. Firstly, in the double pendulum example, some prior knowledge guiding this the discovery includes **(1)** the two terms $\dot{\omega}_2\cos(\Delta\theta)$ for $\dot{\omega}_1$ and $\dot{\omega}_1\cos(\Delta\theta)$ for $\dot{\omega}_2$, which can be easily derived based on our prior understanding of the physical system, are assumed known in the two governing equations, **(2)** the remaining of the formulas are potentially comprised of the free combination of $\dot{\omega}_1$, $\dot{\omega}_2$, $\omega_1$, $\omega_2$, as well as the angles ($\theta_1$, $\theta_2$, $\Delta\theta$) under the trigonometric functions $\cos(\cdot)$ and $\sin(\cdot)$, and **(3)** velocities and relative velocities of two masses, as well as their directions (sign function) might contribute to the damping. Based on the above information, we then design the candidate production rules. The prior knowledge can be easily incorporated in the proposed SPL machine to improve the discovery performance, rather than relying on free combination of mathematical operators and symbols.
>
> Since the double pendulum is a highly chaotic system whose governing equations are very complicated, none of the baseline models could work (e.g., NGGP and GP). Without imposing the prior knowledge in production rule design, although the SPL machine could still produce high-quality result given a large number of episode search, the success rate deteriorates. We believe the use of prior knowledge should be encouraged in practical symbolic regression tasks when dealing with real-world physical systems.
>
> **7. Typos:** Thanks for your careful reading. These typos have been fixed in the revised paper. Note that the phrase of “optimistic policy”, commonly used for reinforcement learning in literature, is correct. In addition, we have conducted another round of careful proofreading of the manuscript for quality improvement.
>
> We hope this response would help clarify the reviewer's concern. Please do not hesitate to let us know if you have any further questions. Thank you very much.

---

> ### Author Response · Authors · 2022-11-17
> **Response to Reviewer X2Dz (Part 1)**
>
> We sincerely thank the reviewer for the positive and constructive comments/suggestions, which are very helpful for improving our paper. Please find our responses below. Revisions have also been made in the paper.
>
> **1. Comparison with vanilla MCTS:** Thanks for this suggestion. We have performed a study of comparison between the proposed SPL machine and vanilla MCTS (also termed as Model D in the ablation study). Please see our reply to the next question.
>
> **2. Ablation Results:** We sincerely thank the reviewer for this important comment. Firstly, we would like to mention that we made a small mistake previously when calculating the average recovery rates for the ablated models. We have now fixed it and conducted the ablation study again. In particular, we consider four ablation studies by removing: (a) the adaptive scaling in reward calculation, (b) the discount factor $\eta^n$ that drives equation parsimony in Eq. 2, (c) module transplantation in tree generation, and (d) all of the above. The resulting models are denoted with Model A, Model B, Model C, and Model D (note that Model D is equivalent to the vanilla MCTS). The ablation study is performed on the 12 classic Nguyen's benchmark problems, where the recovery rate of each model is calculated.
>
> The results of the ablation study are summarized below (see Appendix Table C.1 in the revised manuscript). Results show that the average recovery rates for these models are all smaller than that produced by SPL, where the module transplantation brings the largest effect. In particular, all the ablated models fail to uncover the Nguyen-12 equation (the most difficult case), where the coefficient “1/2” needs to be represented by mathematical operators and symbols, e.g., $x/(x + x)$, $y/(y + y)$, etc.
>
> **Table. Summary of the ablation study results**
>
> | Benchmark | Expression | SPL (%) | Model A (%) | Model B (%) | Model C (%) | Model D (%) |
> |---|:---:|:---:|:---:|:---:|:---:|:---:|
> | Nguyen-1 | $x^3+x^2+x$ | 100 | 100 | 100 | 14 | 12 |
> | Nguyen-2 | $x^4+x^3+x^2+x$ | 100 | 100 | 100 | 0 | 0 |
> | Nguyen-3 | $x^5+x^4+x^3+x^2+x$ | 100 | 100 | 100 | 0 | 0 |
> | Nguyen-4 | $x^6+x^5+x^4+x^3+x^2+x$ | 99 | 96 | 92 | 0 | 0 |
> | Nguyen-5 | $\sin(x^2)\cos(x)-1$ | 95 | 95 | 92 | 92 | 88 |
> | Nguyen-6 | $\sin(x^2)+\sin(x+x^2)$ | 100 | 100 | 96 | 100 | 100 |
> | Nguyen-7 | $\ln(x+1)+\ln(x^2+1)$ | 100 | 100 | 100 | 100 | 100 |
> | Nguyen-8 | $\sqrt{x}$ | 100 | 100 | 100 | 100 | 100 |
> | Nguyen-9 | $\sin(x)+\sin(y^2)$ | 100 | 100 | 100 | 100 | 100 |
> | Nguyen-10 | $2\sin(x)\cos(y)$ | 100 | 100 | 100 | 0 | 0 |
> | Nguyen-11 | $x^y$ | 100 | 96 | 96 | 92 | 87 |
> | Nguyen-12 | $x^4-x^3+\frac{1}{2}y^2-y$ | 100 | 0 | 0 | 0 | 0 |
> |Average | | 93.5 | 90.58 | 89.67 | 49.83 | 48.92 |
>
> **3. Coefficient/Constant Estimation:** Thanks for this question. The constant coefficient value is estimated via concurrently solving an optimization problem, e.g., by the Powell's conjugate direction method (Powel, 1964). The computational cost is high for constant coefficient value due to repeated calls for an optimization process. For example, when the tree structure changes or is updated, the optimization of the constant coefficients should be re-performed simultaneously.
>
> **4. Non-parsimonious Equations:** Thanks for this comment. In fact, the discovered equations for the cases of the yellow whiffle ball and the bowling ball in Appendix Table D.1 are still parsimonious. Note that these formulas listed in the table are the raw equations produced by SRL. Further simplification helps better parsimony of the formulas. For example, the product of exp terms can be merged, the same denominator can be combined, etc.
>
> **5. Importance of Savitzky-Golay Filter:** Thanks for this question. For the nonlinear dynamics example, only the system states (e.g., $x, y, z$ in the Lorenz system) are measured. The derivatives of the system states (e.g., $\dot{x}, \dot{y}, \dot{z}$) are unmeasured but estimated by central difference. When the data has noise or sampled sparsely, numerical errors will be dominant due to the use of finite difference. The Savitzky–Golay filter is helpful to smooth and denoise the derivative estimation reducing the measurement noise effect.

---

### Official Review · Reviewer_GVk6 · 2022-10-25

**Confidence:** 4
**Correctness:** 4
**Technical Novelty And Significance:** 3
**Empirical Novelty And Significance:** Not applicable
**Recommendation:** 8

**Clarity, Quality, Novelty And Reproducibility:**

The paper is high-quality and well-written. I can easily learn the comprehensive review of the existing work and the background of the problem the paper is working on. My only concern is that there might not be enough modeling novelty.


**Strength And Weaknesses:**

**Strengths**

**1. The paper is well-organized and easy to follow.**

The organization and the writing of the whole paper are very attractive to me. The introduction and the background sections go over the different lines of existing work and the background one needs before diving into the proposed method in this paper. Figure 1 and Algorithm 1 clearly illustrate the high-level architecture and the underlying details of the method, respectively. Sections 4 to 7 discuss how the method could be applied to different benchmarks and tasks.

**2. The proposed method adopts three useful adjustments to the conventional MCTS.**

 The proposed Symbolic Physics Learner (SPL) machine carefully incorporates the three adjustments to the conventional MCTS, which largely improve the prediction performance and stability.



**3. The experimental sections are impressive to me.**

The experiments are conducted on four separate cases. The results show a consistent improvement on a bunch of tasks for SPL.


**Weaknesses**

**1. The model novelty.**

I am a little bit concerned about the novelty of the proposed method. It looks like an incremental model that makes several modifications on top of a popular model.


**Summary Of The Paper:**

The paper comprehensively reviews the different families of methods to tackle the data-driven discovery of nonlinear dynamics by comparing their advantages and disadvantages. Based on these discussions, the papers propose a new model, the Symbolic Physics Learner (SPL) machine, empowered by MCTS, to discover the mathematical structure of nonlinear dynamics. The paper proposes three changes: achieving a new expected reward, using adaptive scaling in policy evaluation, and transplanting modules with high returns. SPL shows its effectiveness in a wide range of tasks for discovering the mathematical structure of nonlinear dynamics.


**Summary Of The Review:**

The paper is attractive to me since there are various strengths in terms of problem formulation, modeling, experiments, and writing.

---

> ### Author Response · Authors · 2022-11-17
> **Response to Reviewer GVk6**
>
> We sincerely thank the reviewer for the positive feedback. Please find our responses below. Revisions have also been made in the paper.
>
> **1. Novelty:** Thanks for this comment. We would like to clarify that the novel contributions of this works are two-fold: **(1)** We have made key adjustments to the MCTS, including the greedy search, the adaptive-scaled rewarding, the reward regularizer, and the expression tree module transplantation, to make it possible to efficiently uncover the best path to formulate the complex equations; and **(2)** The SPL machine straightforwardly accepts our prior or domain knowledge, or any sorts of constraints of the tasks in the grammar design while leveraging great flexibility in expression formulation. Please note that our objective is not to improve the MCTS algorithm itself, but to propose an effective approach for both symbolic regression and governing equation discovery tasks. The numerical experiments show that the proposed SPL machine shows great potential along this horizon, which outperforms the SOTA models including NGGP, pySINDy, and GP.
>
> We hope this response would help clarify the reviewer's concern. Please do not hesitate to let us know if you have any further questions. Thank you very much.

---

### Official Review · Reviewer_k15E · 2022-10-25

**Confidence:** 3
**Correctness:** 2
**Technical Novelty And Significance:** 2
**Empirical Novelty And Significance:** 2
**Recommendation:** 8

**Clarity, Quality, Novelty And Reproducibility:**

Quality:
The contents in the manuscript is not reliable due to the low quality of the manuscript.
The presence of even the following elementary typos in the ABSTRACTION would suggest that the paper is not complete.

In abst:
PSL -> SPL

In page 3:
tre -> the

an an -> an

In page 4:
$n$ -> $\eta$

In page 5:
unpredicTable -> unpredictable

In page 7:
DSR -> (The abbreviation is used without declaration.)

Nguyen-12 in table 1:
$x^4-x^2+\frac{1}{2}y^2-y$ -> $x^4-x^3+\frac{1}{2}y^2-y$ (Please check the Table 1 of [Mundhenk et al., 2021])

Clarity:
It is unclear from the demonstrated numerical experiments what aspects of the proposed method are intended to demonstrate its effectiveness.

Originality:
The proposed method is a trivial improvement of the previous method, and the results obtained are not so different from those of the previous method, so the novelty of the proposed method is weak.

**Strength And Weaknesses:**

strengths:
The strength of the research is its stable symbolic regression compared to prior methods.

weaknesses:
The following three points are drawbacks
1 There is no discussion or theoretical analysis of why the proposed method is effective. Therefore, I cannot eliminate the suspicion that the effectiveness of the proposed method over the prior methods depends on the superiority of the hyperparameter tuning.
I think it is good to show what kind of "prior physics knowledge in nonlinear dynamics" can be implemented by the proposed algorithm, and provide numerical experiments to demonstrate its effectiveness.

2 Where the stability of the estimation results due to hyperparameter settings is unclear.
Please explain the optimization method for hyperparameters and the stability of the estimation results with respect to variations in hyperparameters.

3 Where it is unclear whether the hyperparameter tuning of the prior method is appropriate.
It is same as above point. In particular, we would like to know why the results in Table 1 do not match the results of similar experiments presented in the paper proposing the prior method. (Table 1 in [Mundhenk et al., 2021])


**Summary Of The Paper:**

This manuscript proposes a novel symbolic physical learner (SPL) that extracts analytical formulas governing nonlinear dynamics from limited data. The SPL uses a Monte Carlo tree search (MCTS) agent to search for optimal expression trees based on data.
The specific differences of the SPL with respect to previous studies are the following three points. (1) replacing the expected reward in UCT scores with a maximum reward to better fit the equation discovery objective, (2) employing an adaptive scaling in policy evaluation which would eliminate the uncertainty of the reward value range owing to the unknown error of the system state derivatives, and (3) transplanting modules with high returns to the subsequent search as a single leaf node.
The effectiveness and superiority of the SPL machine are demonstrated by numerical experiments comparing it to a baseline of state-of-the-art symbolic regression algorithms.

**Summary Of The Review:**

Symbolic regression is an important technique for future physics research using machine learning. In addition, equation tree search using reinforcement learning, which was improved in this study, is an important technique that has shown its usefulness in other fields as well. As such, this research is important. On the other hand, the description and organization of the paper and the setting of numerical experiments are still insufficient, and I think that the paper should be accepted after these points are improved.

---

> ### Author Response · Authors · 2022-11-17
> **Response to Reviewer k15E (Part 2)**
>
> **6. Clarity and Novelty:** Thanks for this comment. Firstly, we would like to clarify that the novel contributions of this works are two-fold: **(1)** We have made key adjustments to the MCTS, including the greedy search, the adaptive-scaled rewarding, the reward regularizer, and the expression tree module transplantation, to make it possible to efficiently uncover the best path to formulate the complex equations; and **(2)** The SPL machine straightforwardly accepts our prior or domain knowledge, or any sorts of constraints of the tasks in the grammar design while leveraging great flexibility in expression formulation. Secondly, the numerical experiments (especially the nonlinear dynamics discovery) show that the proposed SPL outperforms the SOTA models including NGGP, pySINDy, and GP. Please note that our objective is not to improve the MCTS algorithm itself, but to propose an effective approach for both symbolic regression and governing equation discovery tasks. The proposed SPL machine shows great potential along this horizon.
>
> We hope the revision would address the reviewer's comments. Please do not hesitate to let us know if you have any further questions. Thank you very much.

---

> ### Author Response · Authors · 2022-11-17
> **Response to Reviewer k15E (Part 1)**
>
> We sincerely thank the reviewer for the constructive comments/suggestions, which are very helpful for improving our paper. Please find our responses below. Revisions have also been made in the paper.
>
> **1. Theoretical Analysis:** Thanks for this comment. Since the search engine we employed in our proposed SPL machine is the UCT-based MCTS, the theoretical analysis (e.g.,  convergence, guarantees)) of this algorithm is found in literature such as Shah et al. (2019). We have clarified this in our revised manuscript (see Section 2).
>
> Nevertheless, we have followed the reviewer’s suggestion and added a paragraph discussing why the proposed method is effective in Appendix Section F in our revised manuscript: *The effectiveness of the proposed SPL machine is empowered by the following elements: (1) The use of MCTS enables the flexible representation of search space with customized computational grammars, composed of a finite set of mathematical operators and symbols, to guide the search tree expansion. (2) The exploration-exploitation trade-off nature of MCTS is remarkably useful for search of the optimal mathematical expression tree. (3) The key adjustments, including the greedy search, the adaptive-scaled rewarding, the reward regularizer, and the expression tree module transplantation, make it possible to efficiently uncover the best path to formulate the complex equations. (4) The SPL machine straightforwardly accepts our prior or domain knowledge, or any sorts of constraints of the tasks in the grammar design while leveraging great flexibility in expression formulation.*
>
> **2. hyperparameter settings:** Thanks for this comment. We must acknowledge that the hyper-parameters are empirically selected. However, there are some heuristics behind. For example, $\eta$ (the regularizing parameter which is very close to 1) depends on our desire for parsimony of the target equation, e.g., a smaller value drives the algorithm to search for more parsimonious formula. Typically, we set it to be 0.99$\sim$1. Other hyperparameters (e.g., Maximum Module Transplantation, Episodes Between Module Transplantation, etc.) that control the method convergence are selected empirically. We have performed several parametric trial-and-error tests in the warm-up phase to determine the hyperparameter values that lead to good convergence (see Appendix Table B.2 for example). We added a section (Appendix Section A) in the revised manuscript discussing the hyperparameter setting for the proposed SPL machine.
>
> **3. Prior Physics Knowledge in Nonlinear Dynamics:** This is an excellent remark. The SPL machine straightforwardly accepts our prior or domain knowledge, or any sorts of constraints of the tasks in the grammar design while leveraging great flexibility in expression formulation. For example, in the double pendulum example, some prior knowledge guiding this the discovery includes **(1)** the two terms $\dot{\omega}_2\cos(\Delta\theta)$ for $\dot{\omega}_1$ and $\dot{\omega}_1\cos(\Delta\theta)$ for $\dot{\omega}_2$, which can be easily derived based on our prior understanding of the physical system, are assumed known in the two governing equations, **(2)** the remaining of the formulas are potentially comprised of the free combination of $\dot{\omega}_1$, $\dot{\omega}_2$, $\omega_1$, $\omega_2$, as well as the angles ($\theta_1$, $\theta_2$, $\Delta\theta$) under the trigonometric functions $\cos(\cdot)$ and $\sin(\cdot)$, and **(3)** velocities and relative velocities of two masses, as well as their directions (sign function) might contribute to the damping. Based on the above information, we then design the candidate production rules. The prior knowledge can be easily incorporated in the proposed SPL machine to improve the discovery performance, rather than relying on free combination of mathematical operators and symbols. This has been discussed in Appendix Section E.2 in the revised manuscript.
>
> **4. Results in Table 1:** Thanks for this question. We noticed that the NGGP results reported in Table 1 in our manuscript has slight discrepancy compared with those by Mundhenk et al. (2021) when performing the baseline comparison test. Since we need to compare the training time between SPL and NGGP (see Appendix Table E.3), we re-ran the source code of NGGP on our computer while keeping the hyperparameters the same as those used in the original paper. Given the stochastic nature of the NGGP, the results show very slight difference. We believe this is reasonable.
>
> **5. Quality and Typos:** Thanks for the reviewer’s careful reading. We have fixed the typos. In addition, we have conducted another round of careful proofreading of the manuscript and hope the revised version clears the reviewer’s concern on quality.

---

> ### Comment · Reviewer_k15E · 2022-11-24
> **Thank you for your reply**
>
> Thank you for your thoughtful response.
> My concerns have mostly been answered, so I change my rating.
> However, please check the typo of the manuscript strictly.

---

> > ### Author Response · Authors · 2022-11-24
> > **Thank you for your positive feedback**
> >
> > Dear Reviewer k15E:
> >
> > Many thanks for your kind reply and positive feedback. We sincerely appreciate that you raised the score. We will do another round of proofreading to make sure the typos are fully eliminated.
> >
> > Best regards,
> >
> > *The Authors of the Paper*

---

### Author Response · Authors · 2022-11-17
**Revised paper uploaded and response to reviewers’ comments posted**

Dear Reviewers:

We would like to thank you for your positive and constructive comments, which are very helpful for improving our paper. We have posted point-to-point reply to each question/comment raised by you and uploaded the revised version of our paper (with track changes marked in red). Please do feel free to let us know if you have any further questions.

Thank you very much.

Best regards,

*The Authors of the Paper*

---

### Decision · Program_Chairs · 2023-01-20

**Decision:**

Accept: notable-top-5%

**Justification For Why Not Higher Score:**

The paper is unanimously vetted by the reviewers.

**Justification For Why Not Lower Score:**

The paper is unanimously vetted by the reviewers.

**Metareview: Summary, Strengths And Weaknesses:**

 The method proposes the use of Monte Carlo Tree Search to discover more accurately and robustly symbolic expressions for the observed data from nonlinear dynamical systems.

All reviewers agree on the novelty, as well as the relevance of the empirical findings.

**Note From Pc:**

if the above contains the word "oral" or "spotlight" please see: "oral" presentation means -> notable-top-5% and "spotlight" means -> notable-top-25%. As stated in our emails, we are disassociating presentation type from AC recommendations